# Effects of reciprocal inhibition and whole-body relaxation on persistent inward currents estimated by two different methods

Ricardo N. O. Mesquita[1,2] , Janet L. Taylor[1,2] , Gabriel S. Trajano[3] , Jakob Škarabot[4] , Aleš Holobar[5] , Basílio A. M. Gonçalves[6] and Anthony J. Blazevich[1]

[1]*Centre for Human Performance, School of Medical and Health Sciences, Edith Cowan University, Perth, Australia*
[2]*Neuroscience Research Australia, Sydney, Australia*
[3]*School of Exercise and Nutrition Sciences and Institute of Health and Biomedical Innovation, Queensland University of Technology, Brisbane, Australia*
[4]*School of Sport, Exercise and Health Sciences, Loughborough University, Leicestershire, UK*
[5]*Faculty of Electrical Engineering and Computer Science, University of Maribor, Maribor, Slovenia*
[6]*Griffith Centre of Biomedical and Rehabilitation Engineering (GCORE), Menzies Health Institute Queensland, Griffith University, Brisbane, Australia*

Edited by: Richard Carson & Jing-Ning Zhu

Linked articles: This article is highlighted in a Journal Club article by Thorstensen. To read this article, visit https://doi.org/10.1113/JP283249.

The peer review history is available in the Supporting information section of this article (https://doi.org/10.1113/JP282765#support-information-section).

**Ricardo Mesquita** has completed an undergraduate degree in Sports Science (Faculdade de Motricidade Humana, Portugal), a master's degree in Biology of Physical Activity (University of Jyväskylä, Finland), and the work described herein is part of his PhD in Exercise Science (Edith Cowan University, Australia). Ricardo's PhD has focused on alpha motoneurons – nerve cells in the spinal cord that innervate muscle fibres to produce muscle contractions. With a background in human physiology and neuromuscular control, Ricardo aims to better understand human motor commands and disseminate research findings that can ultimately be used to improve muscle control in healthy and clinical populations.

**Abstract** Persistent inward currents (PICs) are crucial for initiation, acceleration, and maintenance of motoneuron firing. As PICs are highly sensitive to synaptic inhibition and facilitated by serotonin and noradrenaline, we hypothesised that both reciprocal inhibition (RI) induced by antagonist nerve stimulation and whole-body relaxation (WBR) would reduce PICs in humans. To test this, we estimated PICs using the well-established paired motor unit (MU) technique. High-density surface electromyograms were recorded from gastrocnemius medialis during voluntary, isometric 20-s ramp, plantarflexor contractions and decomposed into MU discharges to calculate delta frequency ($\Delta F$). Moreover, another technique (VibStim), which evokes involuntary contractions proposed to result from PIC activation, was used. Plantarflexion torque and soleus activity were recorded during 33-s Achilles tendon vibration and simultaneous 20-Hz bouts of neuromuscular electrical stimulation (NMES) of triceps surae. $\Delta F$ was decreased by RI ($n = 15$, 5 females) and WBR ($n = 15$, 7 females). In VibStim, torque during vibration at the end of NMES and sustained post-vibration torque were reduced by WBR ($n = 19$, 10 females), while other variables remained unchanged. All VibStim variables remained unaltered in RI ($n = 20$, 10 females). Analysis of multiple human MUs in this study demonstrates the ability of local, focused inhibition to attenuate the effects of PICs on motoneuron output during voluntary motor control. Moreover, it shows the potential to reduce PICs through non-pharmacological, neuromodulatory interventions such as WBR. The absence of a consistent effect in VibStim might be explained by a floor effect resulting from low-magnitude involuntary torque combined with the negative effects of the interventions.

(Received 20 December 2021; accepted after revision 13 April 2022; first published online 18 April 2022)

**Corresponding author** Ricardo N. O. Mesquita: Centre for Human Performance, School of Medical and Health Sciences, Edith Cowan University, Perth, Australia.    Email: r.mesquita@ecu.edu.au

**Abstract figure legend** Motoneuron firing partly depends on their intrinsic properties, such as the strength of persistent inward currents (PICs) of calcium ($Ca^{2+}$) and sodium ($Na^{+}$) ions. PICs make motoneurons more responsive to excitatory input. The contribution of PICs to human motoneuron firing is reduced during both reciprocal inhibitions induced by electrical stimulation of the antagonist nerve (left) and whole-body relaxation (right; participants were asked to relax while listening to a soothing piano song).

## Key points

- Spinal motoneurons transmit signals to skeletal muscles to regulate their contraction. Motoneuron firing partly depends on their intrinsic properties such as the strength of persistent (long-lasting) inward currents (PICs) that make motoneurons more responsive to excitatory input.
- In this study, we demonstrate that both reciprocal inhibition onto motoneurons and whole-body relaxation reduce the contribution of PICs to human motoneuron firing. This was observed through analysis of the firing of single motor units during voluntary contractions.
- However, an alternative technique that involves tendon vibration and neuromuscular electrical stimulation to evoke involuntary contractions showed less effect. Thus, it remains unclear whether this alternative technique can be used to estimate PICs under all physiological conditions.
- These results improve our understanding of the mechanisms of PIC depression in human motoneurons. Potentially, non-pharmacological interventions such as electrical stimulation or relaxation could attenuate unwanted PIC-induced muscle contractions in conditions characterised by motoneuron hyperexcitability.

## Introduction

The number and firing rates of spinal alpha motoneurons actively contributing to a muscle contraction strongly influence both the temporal and amplitude characteristics of force production. These firing characteristics are determined by the organisation of ionotropic synaptic input (received from supraspinal and reflex pathways) as well as complex factors that introduce non-linearity in the motoneuron synaptic input-output relationship

(Heckman, Johnson et al., 2008). This non-linearity is primarily driven by persistent inward currents (PICs) generated via fast-activating voltage-gated sodium and slow-activating calcium channels, mostly located at the motoneuron dendrites (Hounsgaard & Kiehn, 1993; Lee & Heckman, 1996). Once activated, PICs amplify and prolong synaptic input (Heckman et al., 2005) in proportion to the levels of serotonin and noradrenaline (i.e. monoamines) at the motoneurons (Lee & Heckman, 1999, 2000). Monoaminergic input is critical to motoneuron firing, with maximal activation of excitatory ionotropic inputs typically producing less than 40% of maximum motor output in the absence of neuromodulation (Heckman, 1994; Heckman, Johnson et al., 2008). PICs both accelerate the initial motoneuron firing and contribute to the repetitive firing required for sustained muscle contraction (Heckman, Johnson et al., 2008), and thus play a critical role in motor control.

Given that PIC channels are voltage-gated and that their activity increases proportionally with monoamine concentrations, PIC strength can be modulated by changing either ionotropic or neuromodulatory input. In fact, PIC activity can be markedly reduced by inhibitory ionotropic synaptic input such as disynaptic Ia reciprocal inhibition, with robust evidence in animal models (Hyngstrom et al., 2007; Kuo et al., 2003) and preliminary observations in humans (Trajano et al., 2014; Vandenberk & Kalmar, 2014). Also, both pharmacological increases and decreases of neuromodulation influence the magnitude of PIC contribution to motoneuron firing (D'Amico et al., 2013; Udina et al., 2010). Despite this, effects of non-pharmacological interventions have yet to be examined. As both arousal state and the level of voluntary activity influence the noradrenergic (Valentino & Van Bockstaele, 2008) and serotonergic (Jacobs et al., 2002) systems, whole-body relaxation might be one intervention expected to strongly reduce monoamine release and thus PIC strength.

Although PIC amplitudes in single motoneurons cannot be directly measured in humans, distinct motor unit (MU) firing patterns that are likely to be generated by PICs have been observed. The paired MU technique, the current standard method to indirectly estimate PIC strength (Gorassini et al., 1998, 2002), quantifies MU recruitment-derecruitment hysteresis during ramped-force muscle contractions. The smoothed firing rate of a lower-threshold MU (control unit) is used as a proxy for the level of net synaptic input at the time of recruitment and derecruitment of a higher-threshold MU (test unit) and the difference in estimated recruitment and derecruitment inputs constitutes the $\Delta F$ (change in frequency) value. Initially validated in animal models (Bennett, Li, Harvey et al., 2001), $\Delta F$ provides an estimation of the contribution of PICs to MU firing. In humans, this method requires invasive intramuscular

electromyography (EMG; e.g. Foley & Kalmar, 2019; Marchand-Pauvert et al., 2019; Wilson et al., 2015) or high-density surface EMG electrode arrays (e.g. Hassan et al., 2021; Khurram et al., 2021; Orssatto et al., 2021) to detect single MU firings using decomposition techniques. These approaches exploit the 1-to-1 ratio between the firing rate of a motoneuron and MU action potentials in the muscle, providing a unique window into the central nervous system, but require computational procedures that are relatively time-consuming and complex. The method also involves voluntary muscle activation, which may not be feasible across all populations (e.g. in complete spinal cord injury) and does not allow PIC estimation in the absence of descending voluntary drive. Thus, other techniques that overcome some of these limitations are of interest.

A potential alternative, or supplementary, method to estimate PIC activity is ongoing tendon vibration overlaid by short bursts of neuromuscular electrical stimulation (NMES) in an otherwise-inactive target muscle (Bochkezanian et al., 2018; Espeit et al., 2021; Kirk et al., 2019; Magalhães & Kohn, 2010; Mesquita et al., 2021; Trajano et al., 2014). This method induces increasing involuntary muscle contraction forces with magnitudes that are larger than would be expected from the direct activation of motor axons. The contribution of peripheral (i.e. within the muscle) mechanisms to this phenomenon is still a matter of debate (Frigon et al., 2011); however, there is increasing support for centrally mediated mechanisms, as recently discussed (Mesquita et al., 2021). Importantly, the evoked, ongoing contractions display characteristics consistent with PIC behaviour. Self-sustained torque has been observed after cessation of both the vibration and NMES, which may be explained by PIC-related bistable behaviour in some spinal motoneurons (Lee & Heckman, 1998), and the progressive increase in torque during the trials (Kirk et al., 2019; Magalhães & Kohn, 2010; Mesquita et al., 2021; Trajano et al., 2014) is consistent with the calcium-dependent facilitation (i.e. warm-up) effect seen during repetitive activation when PIC amplitude has been measured directly (Svirskis & Hounsgaard, 1997). Additionally, the magnitude of involuntary force and muscle activity is muscle length dependent (Trajano et al., 2014), consistent with the effect of reciprocal inhibition observed *in vivo* using voltage clamp (Hyngstrom et al., 2007). Finally, reductions in both ongoing, involuntary plantar flexion torque (Trajano et al., 2014) and soleus $\Delta F$ values (Trajano et al., 2020) have been observed after brief (5 min) muscle stretching, indicating that both techniques provide similar information, at least in some cases. Thus, a technique involving tendon vibration superimposed with neuromuscular electrical stimulation (NMES) bursts, hereafter abbreviated as VibStim (vibration with stimulation), may allow indirect

estimation of PIC activity *in vivo* in the absence of voluntary drive to the muscle using equipment that is available in many laboratories and clinical environments. However, a comparison of this technique with the well-established paired MU technique has yet to be conducted.

Besides a better understanding of the capabilities and limitations of methods to non-invasively estimate PICs in humans, investigation of the effects of different interventions with these two methods was expected to provide insight into the effects of PICs on motoneuron firing in different contexts. Thus, the aim of the present study was to examine the effects of both reciprocal inhibition and whole-body relaxation on (1) the contribution of PIC activity to MU firing in plantar flexor motoneurons, estimated using the paired MU technique, and (2) the magnitude of ongoing, involuntary plantar flexion torque and muscle activity assessed during and immediately after simultaneous tendon vibration and NMES application (VibStim), which is assumed to be proportional to PIC activation. The effect of sex on PIC strength estimated using the two techniques was also examined, for the first time, in an exploratory analysis. It was hypothesised that PIC strength estimated by both techniques would decrease both with reciprocal inhibition, given that PICs are strongly reduced by inhibitory inputs, and in whole-body relaxation, speculatively as a result of a decreased serotonergic and noradrenergic release onto motoneurons associated with the reduced muscle activity, global stress levels, and arousal.

## Methods

This research formed part of a larger study assessing the acute effects of multiple interventions on the estimated contribution of PIC activity to motoneuron firing (paired MU technique) and magnitude of evoked involuntary torque during application of tendon vibration with NMES (VibStim). Data relating to the effects of reciprocal inhibition and whole-body relaxation will be presented herein (see Procedures for details).

### Participants and ethical approval

Twenty-one healthy adults aged 19−36 years (11 males and 10 females; age: 26.3 ± 5.1 years; body mass: 79.4 ± 15.6 kg; height: 172.9 ± 10.1 cm) volunteered for the study. Exclusion criteria included diagnosed neurological disorders, current or recent injuries (recovered for less than 6 months) that required clinical assessment, and medications that might affect central monoamine concentrations. Participants were fully informed of any risks or discomforts associated with the procedures before giving their written informed consent to participate. The

procedures were approved by the Human Research Ethics Committee of Edith Cowan University (reference number: 22306), and performed according to the *Declaration of Helsinki*, except for registration in a database.

### Procedures

The participants visited the laboratory on three occasions. The first visit was a familiarisation session during which participants were accustomed to maximal plantar flexion strength assessment, ramp contractions, and VibStim. Participants returned to the laboratory at the same time of the day on two experimental days separated by at least 24 h (mean = 3.0 ± 1.9 days apart). Participants were asked to abstain from caffeine-containing foods and beverages within 12 h of the sessions.

In all sessions, participants sat in the chair of an isokinetic dynamometer (Biodex System 4, Biodex Medical System, USA), with the hips flexed to 50° (0° = extended neutral position), right knee fully extended (0°) and right ankle at 10° of dorsiflexion (0° = anatomical position). The right foot was tightly fixed to the plantar flexor attachment and the axis of rotation aligned with the lateral malleolus.

On experimental days, VibStim and paired MU techniques were performed under four experimental conditions. The four conditions were performed in a randomised order over two experimental days with two conditions per day. However, VibStim was always performed first as pilot testing revealed that skin preparation procedures for high-density surface electromyography (HD-sEMG) increased participant discomfort during NMES. Thus, participants completed two conditions with VibStim and then two conditions with the paired MU technique.

Before testing, NMES electrodes (self-adhesive; 5 × 9 cm; Dura-Stick Plus, DJO Global, USA) were placed over the triceps surae muscles with the cathode placed transversely and distal to the popliteal crease and the anode transversely over the distal gastrocnemius-Achilles muscle-tendon junction. Bipolar electrodes to measure electromyographic (EMG) signals were also placed over soleus on both days. Participants then performed four voluntary submaximal isometric plantar flexion contractions (∼3-s contractions at 20, 40, 60 and 80% of perceived maximal effort) as warm-up, then three maximal voluntary contractions (MVC) with a 90-s intertrial rest. After a 3-min rest period, 0.5-s 20-Hz trains of NMES were applied percutaneously to the plantar flexors at increasing intensities using a constant-current stimulator (DS7, Digitimer Ltd, UK) to determine the current required to evoke 20% of MVC torque (the maximum value achieved during the familiarisation session was used).

During testing, control trials were always performed before the trials with the experimental interventions (i.e. for each condition: 2 control + 2 experimental VibStim trials, and 3 control + 3 experimental trials with the paired MU technique). A 90-s rest was provided between all trials, and ~20 min separated tests using the different techniques, during which skin preparation and HD-sEMG electrode placement were completed. A permanent marker was used to mark electrode locations to ensure consistent between-session electrode placement. As the data were collected as part of a larger study, data relating to the effects of interventions in which we hypothesised a decrease in PIC activity (RI and WBR) will be presented herein. The other interventions were jaw clenching and a countback task in intervals of 13, in which we expected an increase of PICs. At the end of the session on the day in which the RI condition was performed, stimuli were also delivered to the common peroneal nerve (CPN) during 5% MVC plantar flexion to examine the magnitude of the reciprocal inhibition from the ankle dorsiflexors onto the ankle plantar flexors.

### Paired motor unit technique

Isometric 20-s ramp plantar flexion contractions to 20% of MVC (assessed during the familiarisation session) were performed with an 8-s ascending phase, 4-s hold phase and 8-s descending phase (contraction/relaxation rate = 2.5% MVC/s). Real-time visual feedback of plantar flexion torque signals from the isokinetic dynamometer allowed participants to follow a torque path on a large screen (ISO-AUXSE adapter and OT Biolab+ software, OT Bioelettronica, Italy).

HD-sEMG was recorded from gastrocnemius medialis (GM) using a flexible adhesive grid of 32 equally spaced electrodes (GR10MM0804, 10-mm inter-electrode distance; OT Bioelettronica). The skin under the electrodes was shaved, abraded with sandpaper, and swabbed with alcohol. The grids were attached to the skin (distal region of GM) by bi-adhesive foam with conductive cream filling the adhesive foam wells (AC Cream, Spes Medica, Italy). A damp strap electrode (WS2, OT Bioelettronica, Italy) around the ankle was used as a reference electrode. EMG signals were amplified ($256\times$), recorded in monopolar mode after digital conversion at 2000 Hz, and band-pass filtered (10–500 Hz) by a 16-bit wireless amplifier (Sessantaquattro, OTBioelettronica). EMG signals were recorded and visualised using OTBiolab+ software (version 1.3.0.0, OT Bioelettronica).

**Data analysis – paired motor unit technique.** The paired MU technique (Gorassini et al., 1998, 2002) was used to estimate PICs. The smoothed firing rate of a lower threshold MU (control unit) was used to estimate the level of net synaptic input at recruitment and derecruitment of a higher-threshold MU (test unit). The difference between the recruitment and derecruitment inputs constituted the $\Delta F$ (change in frequency) value, an estimation of the contribution of PICs to motoneuron firing of the test unit (Fig. 1B).

Data files were processed offline with the DEMUSE tool (version 5.0, The University of Maribor, Slovenia) for EMG decomposition via blind source separation with Convolution Kernel Compensation (Holobar & Zazula, 2007) and then MU tracking and manual inspection of spike trains were conducted before further analysis. Band-pass zero-phase (20–500 Hz), zero-phase $2^{nd}$ order finite impulse response high-pass differential (230 Hz), and notch (50 Hz and its higher harmonics) filters were applied in DEMUSE software, and two channels with the lowest signal-to-noise ratio removed to optimise the subsequent decomposition. Thereafter, 50 sequential decomposition runs were conducted in each ramp contraction independently. MU filters (action potential shapes) were identified from each individual ramp contraction and then applied to all the trials of interest (i.e. 3 control trials + 3 experimental trials) through a 2-step semi-automatic tracking approach. This allowed the same MUs to be tracked across different contractions. A quick manual inspection of the spike trains was initially conducted and inaccurate MUs removed (i.e. MUs without a clear identification of firing events from recruitment to derecruitment). If, after this process, 10 or more MUs were identified, a careful inspection of the spike trains (Del Vecchio et al., 2020) was conducted in two of the three ramps in each block (i.e. 2 control ramps + 2 ramps of the experimental condition). The ramps for further analysis were selected by visual inspection. Selection was based on the number of MUs identified, smoothness and adherence to the torque template, and MU firing profiles. If fewer than 10 MUs were identified after the 2-step automatic tracking, the HD-sEMG signals from one ramp were redecomposed. In order to fully exploit the frequency bandwidth of the processed HD-sEMG signals, the notch and high-pass differential filters were not applied in the second decomposition run. Newly identified MUs from the second decomposition run were added to those from the first decomposition run. Filters of the new MUs were manually refined in DEMUSE tool and applied to the other trials, one by one, for tracking purposes.

Electrical stimulation artefacts were present in the HD-sEMG signals during the RI trials, engulfing the EMG. A novel 'offline' approach was therefore used to allow MU firing identification. Briefly, 50 sequential decomposition runs were conducted in each control ramp, as described above, but not in the ramps with the stimulus artefact. Thereafter, to identify MU firings in the RI ramps, MU filters of the control ramps were automatically

assessed and manually refined in the RI ramps, one by one, without applying digital filters.

After carefully editing the spike trains, additional MATLAB (Version 2020b, MathWorks, USA) scripts and functions were used to convert MU firing events into instantaneous firing frequencies and to fit them preferentially with a 5th-order polynomial function. All polynomials were visually inspected and if edge effects were observed at MU recruitment or derecruitment with a 5th-order polynomial (i.e. a clear mismatch between the change in the smoothed and instantaneous firing rate), then a 4th-order polynomial was used. If edge effects were observed for both 5th- and 4th-order polynomials, the MU from that specific trial was not included in further analyses.

A MU pair was only considered for analysis if the test unit was derecruited before the control unit. Furthermore, to identify suitable MU pairs, several criteria were adopted to test the assumption that the control unit was a suitable proxy for net synaptic input. First, test units had to be recruited at least 1 s after the control units (Hassan et al., 2020). Second, the rate-to-rate correlations between the smoothed firing rate polynomials of the test and control units (2000 data points per second) were $r \geq 0.7$ (Stephenson & Maluf, 2011), with the first 500 ms of the test unit being excluded from the correlation analysis (Mottram et al., 2009). Third, a saturation criterion was used to ensure that the control unit increased its firing rate by at least 0.5 Hz after the recruitment of test unit (Stephenson & Maluf, 2011).

A novel two-ramp, multi-control repeated measures approach was used to estimate the contribution of PIC activity to the firing of test units. First, $\Delta F$ values of all possible MU pairs in each ramp were calculated. Then, a merged set of pairs were identified from both ramps in each condition; if a certain pair was identified in both ramps, the average $\Delta F$ was calculated. Finally, in each condition, $\Delta F$ values were calculated for individual test units as the average value obtained when the units were paired with multiple suitable control units, as previously conducted (Trajano et al., 2020). This repeated measures analysis solely included MU pairs that were observed in both conditions, making the subsequent statistical analysis more robust. Onset and offset parameters were quantified using LabChart macros (Version 8.1.16, ADInstruments, New Zealand). The identification of suitable pairs and calculation of $\Delta F$ values was conducted in Excel (Version 2106, Microsoft Corporation, USA).

### Tendon vibration with superimposed NMES (VibStim)

During VibStim tests, tendon vibration was combined with percutaneous NMES to evoke involuntary muscle contractions (Mesquita et al., 2021; Trajano et al., 2014). Achilles tendon vibration was applied at 115 Hz and 1 mm amplitude for 33 s using a hand-held vibrator (Vibrasens, Techno Concepts, France). An experienced researcher held the vibrator with a steady pressure on the posterior aspect of the tendon at the level of the medial malleolus. Ten seconds after vibration onset, $5 \times 2$-s NMES rectangular bursts at 4-s intervals (2 s on, 2 s off) with wide (1 ms) pulse width, 20-Hz frequency, and intensity that evoked 20% of MVC torque were applied to the triceps surae whilst tendon vibration continued (Mesquita et al., 2021). The current required to evoke 20% of MVC torque was re-assessed in each experimental session. During trials, participants held shoulder straps of the chair, looked forwards towards a blank display monitor, remained quiet, did not voluntarily activate their leg muscles, and silently counted backwards from 50 by ones.

Soleus EMG signals were recorded using LabChart Software (ADInstruments, v. 8.1.16, New Zealand) at a 2000-Hz analog-digital conversion rate (PowerLab 16/30, ADInstruments, New Zealand), band-pass filtered (10–1000 Hz), and amplified using a BioAmp EMG system (ADInstruments, New Zealand). After skin preparation, two pre-gelled silver chloride self-adhesive surface electrodes (Blue Sensor N-00-S, 28 mm$^2$, oval shaped, Ambu, Denmark) were placed over the mid-dorsal line of the posterior shank below the gastrocnemius in a bipolar configuration with inter-electrode distance of 1 cm and the reference electrode on the lateral malleolus.

**Data analysis – VibStim.** Four torque-related and three EMG-related variables were calculated (Fig. 1*A*) using LabChart macros. A digital band-pass filter (20–500 Hz) was applied to the EMG signal whereas torque data were filtered using a digital low-pass filter with an 8 Hz cut-off frequency, determined by residual analysis. Torque-related variables are presented as % MVC. Soleus EMG amplitude was quantified as the root mean square (rmsEMG) value in the 500-ms windows of interest (Fig. 1*A*).

The following variables were calculated:

(1) Reflexive torque during vibration ($T_{vib}$) and soleus EMG during vibration ($EMG_{vib}$): mean torque and rmsEMG in a 500-ms window commencing 500 ms after the cessation of NMES, but during tendon vibration.

(2) Self-sustained torque 0.5 s and 3 s post-vibration ($T_{sust0.5}$ and $T_{sust3}$) and soleus EMG at the same times ($EMG_{sust0.5}$ and $EMG_{sust3}$): mean torque in 500-ms windows commencing 500 ms and 3 s after the cessation of tendon vibration and rmsEMG over the same time windows.

(3) Warm-up: the difference between $T_{vib}$ and the mean torque in a 500-ms window commencing 500 ms after the first burst of NMES.

### Interventions (conditions)

**Reciprocal inhibition (RI).** Reciprocal inhibition of the plantar flexors was induced by electrical stimulation of the CPN using a constant-current stimulator (DS7, Digitimer Ltd, UK). Self-adhesive surface electrodes (2.2-cm diameter, Kendall Meditrace 200, Kendall Inter-

national, USA) were placed above the head of the fibula (cathode) and over the ventral branch of the CPN (anode). The anode was positioned so that a tibialis anterior compound muscle action potential (M-wave) was evoked without an M-wave in peroneus longus (Meunier et al., 1993). Bipolar, surface electrodes (Blue Sensor N-00-S, 28 mm$^2$, Ambu, Denmark) were placed on peroneus longus at 1/4 distance between the head of the fibula and lateral malleolus. EMG signals from tibialis anterior were recorded with one electrode at 1/3 of the distance between the lateral epicondyle of the knee and the medial

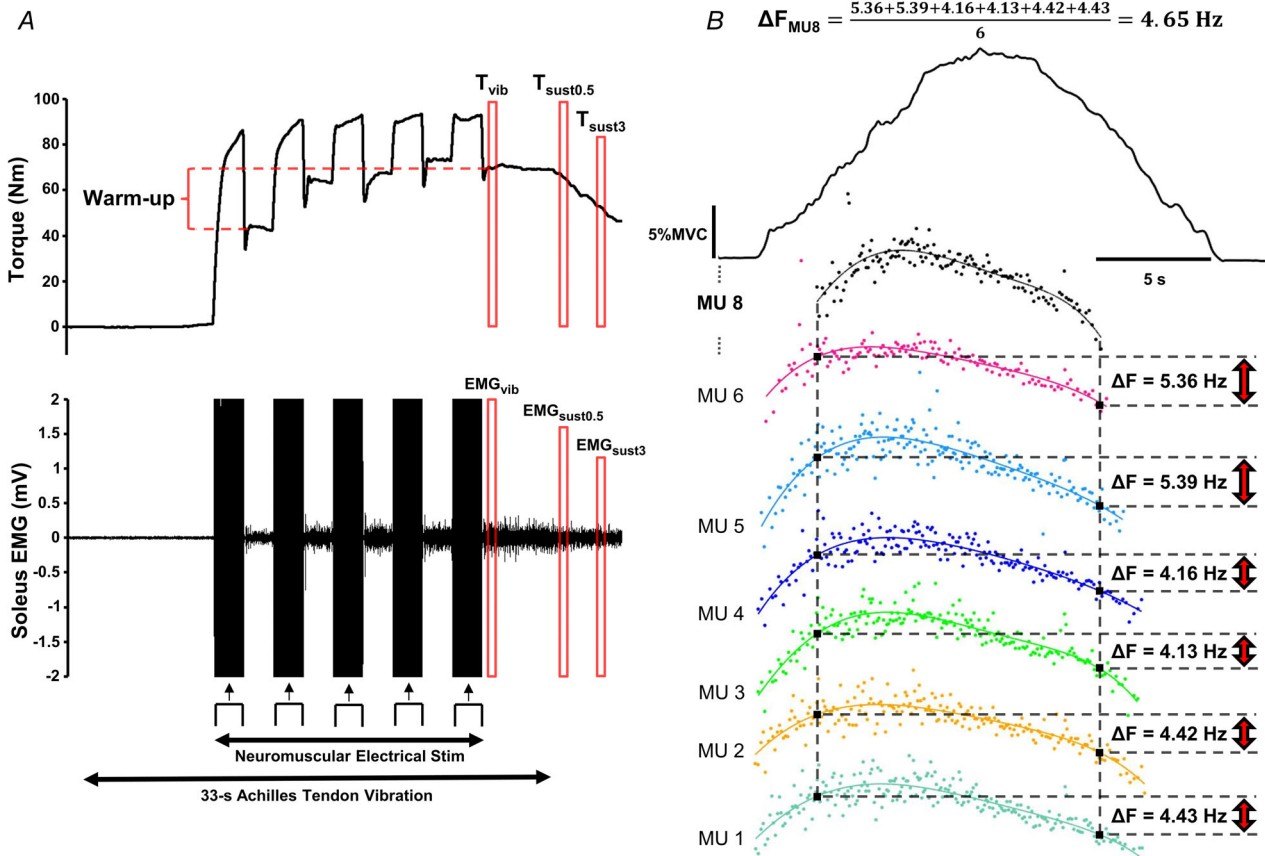

**Figure 1. Data from one participant during single trials using each technique**

*A*, Plantar flexion torque and the soleus EMG response to tendon vibration with superimposed neuromuscular electrical stimulation (VibStim). The Achilles tendon was mechanically vibrated for 33 s, with NMES applied after 10 s of vibration whilst tendon vibration continued. $T_{vib}$: torque measured during vibration, 500 ms after the last burst of NMES; $T_{sust0.5}$ and $T_{sust3}$: torque measured 0.5 and 3 s after vibration cessation, respectively; Warm-up: difference between $T_{vib}$ and torque after the first burst of NMES; $EMG_{vib}$: soleus EMG during vibration, 500 ms after the last burst of NMES; $EMG_{sust0.5}$ and $EMG_{sust3}$: soleus EMG measured 0.5 and 3 s after vibration cessation, respectively. *B*, Calculation of $\Delta F$ from one test unit identified during a ramp contraction in a trial for the paired motor unit (MU) technique. Torque trace (black, at the top) is shown during a 20-s ramp contraction (8-s ascending phase, 4-s hold phase and 8-s descending phase) up to 20% MVC. The instantaneous and smoothed (polynomial fit) firing rate of seven out of 20 MUs that were identified in this trial are shown to demonstrate the paired MU analysis approach. MU 8 (black) was the test unit, which formed suitable pairs with six other control units (MUs 1−6). Thus, $\Delta F$ of MU 8 ($\Delta F_{MU8}$) was the average of $\Delta F$ scores computed when paired with those six control units. Dashed vertical lines mark the time of occurrence of recruitment and derecruitment of the test unit. Dashed horizontal lines indicate the smoothed firing rate of the control units at these times, with the magnitude of difference between these two ($\Delta F$) marked by the red arrows. [Colour figure can be viewed at wileyonlinelibrary.com]

malleolus and over the distal muscle-tendon junction. Bipolar EMG signals from soleus were recorded as previously described and reference electrodes were placed on the malleoli. CPN stimulation (1-ms pulses, 1 Hz) was delivered at an intensity of $1.1 \times$ motor threshold, from ∼10 s before each RI trial until the end of the trial. The imposition of electrical stimuli did not detectibly affect the participant's ability to follow the on-screen force trace, in the ramp contractions. At the end of the session, 100 CPN stimuli (1 Hz) were delivered during 5% MVC plantar flexion. The onset of the inhibitory period was defined as the instant between 35 and 55 ms after stimulation when the rectified EMG values decreased below 2 SD of the mean rectified pre-stimulus values of the averaged waveform, and the offset as the time when the rectified EMG returned to this level. Inhibition magnitude was quantified as the decrease in mean rectified EMG activity during the inhibitory period as a percentage of the background EMG (mean rectified EMG activity between 50 and 10 ms before the stimulation). If an inhibitory period could not be identified for at least 2 ms in the 55-ms time-window post-stimulation, the mean rectified EMG activity between 40 and 50 ms post-stimulation was quantified and expressed as a percentage of the background EMG (adapted from Petersen et al., 1998).

**Whole-body relaxation (WBR).** During WBR, the lights of the laboratory were turned off and participants were asked to deliberately relax all muscles and close their eyes for 30 s before the trial, while listening to a soothing piano song, played through a speaker. Immediately before the trials, participants slowly opened their eyes but kept their muscles relaxed during VibStim and relaxed all muscles except for the plantar flexors during the ramp contractions, whilst the music continued. During the trials, full body support was provided by the chair of the isokinetic dynamometer to decrease the postural demand on the muscles (i.e. participants did not have to hold their head up, like in the other trials).

### Manipulation check

Self-rated stress during the trials was measured immediately after each trial. Participants answered the question 'Please indicate how relaxed or stressed you felt during the trial'. Participants indicated their answer on a seven-point Likert scale, where 1 = very relaxed, 2 = relaxed, 3 = somewhat relaxed, 4 = neither relaxed or stressed, 5 = somewhat stressed, 6 = stressed, and 7 = very stressed. Researchers were blinded to the participants' answers.

During VibStim trials, electrocardiographic (ECG) data were also recorded at 2000 Hz and with an analog band-pass filter (10–120 Hz) to measure heart rate offline. ECG data were not recorded during the paired MU technique trials, as pilot testing demonstrated that ECG electrode connection increased the background noise in HD-sEMG signals. Two electrodes (Blue Sensor N-00-S, 28 mm², Ambu, Denmark) were placed below the left clavicle and on the left lower border of the rib cage, with a reference electrode on the dorsal side of the left hand. ECG data were analysed using the Heart Rate Variability Module in LabChart, which automatically detected R-wave peaks.

### Statistical analysis

Most statistical analyses of MU variables were conducted in R (version 4.0.4) using RStudio environment (version 1.4.1106), except where indicated. Separate repeated measures nested linear mixed-effects models (*lmerTest* package; Kuznetsova et al., 2017) were used to examine the effect of each experimental intervention on $\Delta F$, MU recruitment threshold (i.e. torque at the time when the MU began discharging action potentials), MU derecruitment threshold (i.e. torque at the time when the MU stopped discharging action potentials), and peak smoothed firing rates of the whole sample of test units or MUs (Boccia et al., 2019). Only MUs that could be tracked between control and experimental trials were included in statistical analyses. MU variables were analysed with a random intercepts (parallel slopes) model using 'condition', 'sex' and their interaction as fixed effects, and 'participant' and 'test unit' (or 'MU') as random effects. Residuals were plotted against fitted values to assess whether variance was consistent across the fitted range and Q-Q plot inspection was used to assess the assumption of normality of residuals. In *post hoc* tests, estimated marginal means and their differences (with 95% confidence intervals) were determined using the *emmeans* package (Lenth & Lenth, 2018). For visualisation purposes, kernel density estimations (density curves) of the MU variables were plotted (*gghalves* package; Tiedemann, 2020), to depict data distribution. These density curves are a smooth empirical probability density function, and each data point has an equivalent influence on the final distribution. Exploratory repeated measures correlation coefficients of MU data were computed using the *rmcorr* package (Bakdash & Marusich, 2017). Rate-to-rate Pearson's correlation coefficients between the smoothed firing rate polynomials of the test and control units (2000 data points per second) were computed in Excel (Version 2106, Microsoft Corporation, USA). Independent two-tail *t* tests were used to compare the number of MUs between males and females (IBM SPSS Statistics version 27, SPSS Inc., USA).

Dependent variables obtained during VibStim were analysed using SPSS Statistics software (version 27, SPSS Inc., USA). Data normality was examined using the Shapiro-Wilk test. Torque and EMG variables were

subjected to separate 2 (sex) × 2 (condition: control *vs.* experimental condition) mixed ANOVA. The least significant difference method was used for *post hoc* comparisons, with effect sizes reported as partial eta squared ($\eta_p^2$). An exploratory analysis was also conducted to examine differences with the exclusion of participants who did not exhibit an involuntary evoked torque in the time window of interest during the control condition (i.e. torque <1% MVC). Paired *t* tests were conducted when differences between experimental and control conditions were normally distributed, or alternatively, Wilcoxon signed rank tests were performed.

Additional exploratory Pearson and Spearman's correlations were computed in SPSS Statistics software (version 27, SPSS Inc., USA). Finally, heart rate and self-rated stress were subjected to separate 2 (sex) × 2 (condition: control *vs.* experimental condition) mixed ANOVAs. Statistical significance was set at an alpha level of 0.05 in all statistical analyses.

## Results

### Summary

The main results of the study are summarised in Table 1. The estimate of PIC strength ($\Delta F$) in the paired MU technique was decreased in RI and WBR. Using VibStim, some variables were found to decrease in WBR but generally remained unaltered in RI.

### Descriptive motor unit data

In RI, 889 firing patterns were discriminated across participants whilst 668 firing patterns were discriminated in WBR. 95.8% and 99.0% of MUs were tracked between control and experimental trials in RI and WBR, respectively. The redecomposition of the HD-sEMG signals of participants for whom fewer than 10 MUs were identified resulted in identification of 3.4 ± 2.0 additional MUs in RI and 1.3 ± 1.7 in WBR. For participants with MUs included in statistical analysis, there was no significant difference in the number of MUs per participant between females and males in RI ($P = 0.223$; females: 9.6 ± 7.0, males: 14.0 ± 7.3) or WBR ($P = 0.221$; females: 9.0 ± 3.8, males: 12.3 ± 6.0). There was also no difference in the number of test units in RI ($P = 0.350$; females: 6.6 ± 6.1, males: 9.7 ± 5.7) or WBR ($P = 0.152$; females: 5.0 ± 3.4, males: 8.3 ± 4.7). Other descriptive statistics related to MU data are presented in Table 2.

### Paired motor unit technique

**Reciprocal inhibition.** The estimated marginal means of $\Delta F$ scores in control and RI were 3.37 Hz [3.00, 3.74]

**Table 1. Summary of main results**

| Variable | RI | WBR |
|---|---|---|
| $\Delta F$ | ↓ | ↓ |
| Recruitment threshold | ↔♀, ↓♂ | ↑ |
| Derecruitment threshold | ↔♀, ↓♂ | ↔ |
| Peak smoothed firing rate | ↓♀, ↔♂ | ↓ |
| | | |
| $T_{vib}$ | ↔ | ↓ |
| $EMG_{vib}$ | ↔ | ↔ |
| $T_{sus\,0.5}$ | ↔ | ↓ |
| $EMG_{sust0.5}$ | ↔ | ↔, EDA:↓ |
| $T_{sust3}$ | ↔, EDA:↓ | ↔ |
| $EMG_{sust3}$ | ↔ | ↔ |
| Warm-up | ↔ | ↔ |

Variables quantified during the paired motor unit technique trials are in the upper section of the table and those quantified during VibStim in the lower section. Abbreviations: RI, reciprocal inhibition; WBR, whole-body relaxation; ↓, significant decrease from control to experimental trials; ↑, significant increase from control to experimental trials; ↔, no significant change between control and experimental trials; EDA, exploratory data analysis, excluding participants who did not exhibit an involuntary evoked torque in the time window of interest during the control condition; $\Delta F$, estimate of PIC contribution to MU firing; $T_{vib}$, reflexive torque during vibration; $EMG_{vib}$, soleus EMG during vibration; $T_{sust0.5}$, self-sustained torque 0.5 s post-vibration; $EMG_{sust0.5}$, soleus EMG 0.5 s post-vibration; $T_{sust3}$, self-sustained torque 3 s post-vibration; $EMG_{sust3}$, soleus EMG 3 s post-vibration; Warm-up, torque increase during the trial.

and 3.04 Hz [2.67, 3.41], respectively (Fig. 2*A*). There was a significant [$F_{(1,128)} = 9.41$, $P = 0.003$] ~9.8% decrease in $\Delta F$ values from control to RI (estimated mean difference = −0.33 Hz [−0.55, −0.12]), but there was no significant effect of sex ($P = 0.977$) or interaction between sex and intervention ($P = 0.720$). The recruitment threshold of the test units in control was not statistically associated with the change in $\Delta F$ ($r_{rm} = -0.11$, $P = 0.236$).

There was a significant [$F_{(1,219)} = 9.41$, $P = 0.002$] decrease in smoothed peak firing rate from control to RI (Fig. 2*B*). Moreover, a significant interaction between sex and intervention was detected, with a significant decrease in females (13.4 [11.4, 15.4] to 12.9 Hz [10.9, 14.9], $P < 0.001$, 67 MUs) but no change in males (12.1 [10.5, 13.7] to 12.2 Hz [10.6, 13.8], $P = 0.697$, 154 MUs). The recruitment threshold in control was not statistically associated with the change in peak smoothed firing rate ($r_{rm} < 0.01$, $P = 0.975$).

There was no significant effect of intervention (Fig. 2*C*, $P = 0.758$) or sex ($P = 0.207$) on recruitment threshold. Nonetheless, a significant interaction between sex and intervention was observed [$F_{(1,219)} = 5.90$, $P = 0.016$]. In females, there was no significant change ($P = 0.205$) in

**Table 2. Descriptive statistics for motor unit (MU) data across participants**

| | Reciprocal inhibition | Whole-body relaxation |
|---|---|---|
| Decomposed MUs | 236 | 193 |
| % Polynomials 5th degree | 87.7 | 88.3 |
| % Polynomials 4th degree | 9.8 | 11.7 |
| % Polynomials excluded | 2.5 | 5.4 |
| MUs included for analysis | 221 | 174 |
| Test units | 130 | 101 |
| Pairs | 637 | 391 |
| Participants with MUs for analysis (F/M) | 18 (7/11) | 16 (7/9) |
| Participants with test units for analysis (F/M) | 15 (5/10) | 15 (7/8) |

Decomposed MUs: number of accurate MUs identified and manually inspected. % Polynomials 5th/4th degree: percentage of MUs fitted with a 5th- or 4th-order polynomial function across contractions. % Polynomials excluded: percentage of MUs that could not be fitted with a 5th- or 4th-order polynomial across contractions. MUs included for analysis: number of MUs with a polynomial fit that could be tracked from control to experimental trials, regardless of whether they were included in $\Delta F$ calculations. Test units: number of test units used for analysis and that could be tracked from control to experimental trials. Pairs: number of pairs that could be tracked from control to experimental trials. Participants with MUs for analysis (F/M): number of participants included in recruitment threshold, derecruitment threshold and smoothed peak firing rate (females/males) analyses. Participants with test units for analysis (F/M): number of participants included in $\Delta F$ analysis (females/males).

recruitment threshold from control (5.6% MVC [3.3, 7.9]) to RI (6.0% MVC [3.6, 8.3]), but recruitment threshold decreased ($P = 0.014$) in males from control (7.8% MVC [6.0, 9.6]) to RI (7.4% MVC [5.6, 9.2]). The recruitment threshold in control was weakly associated with the change in recruitment threshold ($r_{rm} = -0.21$, $P = 0.002$).

There was no significant effect of intervention (Fig. 2D, $P = 0.098$) or sex ($P = 0.287$) on derecruitment threshold. Similar to recruitment threshold, a significant sex × intervention interaction was observed [$F_{(1,219)} = 17.48$, $P < 0.001$], with no significant change from control (12.0% MVC [9.9, 14.2]) to RI (12.4% MVC [10.21, 14.6]) detected in females ($P = 0.132$), but a significant decrease from control (11.2% MVC [9.5, 12.9]) to RI (10.4% MVC [8.7, 12.0]) in males ($P < 0.001$). The recruitment threshold in control was not statistically associated with the change in derecruitment threshold ($r_{rm} = -0.09$, $P = 0.218$).

An examination of whether changes in $\Delta F$ were accompanied by changes in other dependent variables (Fig. 3A) revealed: (1) no significant correlation between the change in $\Delta F$ and change in peak smoothed firing rate of test units ($r_{rm} = 0.15$, $P = 0.111$), (2) a weak but significant correlation ($r_{rm} = 0.27$, $P = 0.003$) indicating a greater decrease in $\Delta F$ of test units associated with a greater decrease in their recruitment threshold; and (3) a moderate correlation ($r_{rm} = -0.42$, $P < 0.001$) indicating a greater decrease in $\Delta F$ of test units associated with a greater increase in their derecruitment threshold. Furthermore, $\Delta F$ was not associated with the recruitment threshold of test units in control ($r_{rm} = -0.03$, $P = 0.710$) or RI trials ($r_{rm} = -0.05$, $P = 0.589$).

**Whole-body relaxation.** The estimated marginal means of $\Delta F$ scores in control and WBR were 3.51 Hz [2.99, 4.03] and 3.03 Hz [2.51, 3.54], respectively (Fig. 4A), revealing a significant ~13.7% decrease ($F_{(1,99)} = 16.95$, $P < 0.001$), with a mean difference of $-0.49$ Hz [$-0.72$, $-0.25$]. There was no significant effect of sex ($P = 0.510$) or sex × intervention interaction ($P = 0.263$). The recruitment threshold of test units in control was not statistically associated with the change in $\Delta F$ ($r_{rm} = 0.08$, $P = 0.485$).

There was a significant [$F_{(1,172)} = 9.43$, $P = 0.002$] ~2.3% decrease in the smoothed peak firing rate (Fig. 4B) from control (12.8 Hz [11.4, 14.1]) to WBR (12.5 Hz [11.2, 13.9]), with a mean difference of $-0.26$ Hz [$-0.42$, $-0.09$], but no significant effect of sex ($P = 0.117$) or sex × intervention interaction ($P = 0.738$). The recruitment threshold in control was not statistically associated with the change in peak smoothed firing rate ($r_{rm} = -0.08$, $P = 0.346$).

There was a significant [$F_{(1,172)} = 6.9$, $P = 0.009$] increase in recruitment threshold (Fig. 4C) from control (7.0% MVC [5.9, 8.2]) to WBR (7.6% MVC [6.5, 8.8]), with a mean difference of 0.6% MVC [0.1, 1.0]. There was no effect of sex ($P = 0.482$) or sex × intervention ($P = 0.104$). The recruitment threshold in control was weakly associated with the change in recruitment threshold ($r_{rm} = -0.37$, $P < 0.001$).

The derecruitment threshold (Fig. 4D) showed no significant effect of intervention ($P = 0.252$), sex ($P = 0.817$) or interaction between these two factors ($P = 0.189$). The estimated marginal means of the derecruitment threshold in control and WBR were 11.6% MVC [10.2, 13.0] and 11.3% MVC [9.90, 12.7],

respectively, with a mean difference of $-0.3\%$ MVC $[-0.8, 0.2]$. The recruitment threshold in control was not statistically associated with the change in derecruitment threshold ($r_{rm} = 0.15$, $P = 0.054$).

An examination of whether changes in $\Delta F$ were accompanied by changes in other dependent variables (Fig. 3*B*) revealed: (1) no significant correlation between the change in $\Delta F$ and the change in peak smoothed firing rate of test units ($r_{rm} = -0.01$, $P = 0.900$), (2) no significant correlation between the change in $\Delta F$ and the change in recruitment threshold ($r_{rm} = 0.10$, $P = 0.336$); (3) a weak but significant correlation ($r_{rm} = -0.27$, $P = 0.011$), indicating a greater decrease

in $\Delta F$ of test units associated with a greater increase in their derecruitment threshold. Furthermore, $\Delta F$ was not statistically associated with the recruitment threshold of test units in control ($r_{rm} = -0.06$, $P = 0.552$) or WBR trials ($r_{rm} = 0.01$, $P = 0.917$).

## VibStim

**Reciprocal inhibition.** One participant was excluded from analysis due to voluntary contractions of the leg during the trials, and the EMG-related variables from another participant were excluded due to technical problems (torque variables: $n = 20$, EMG variables: $n = 19$). No

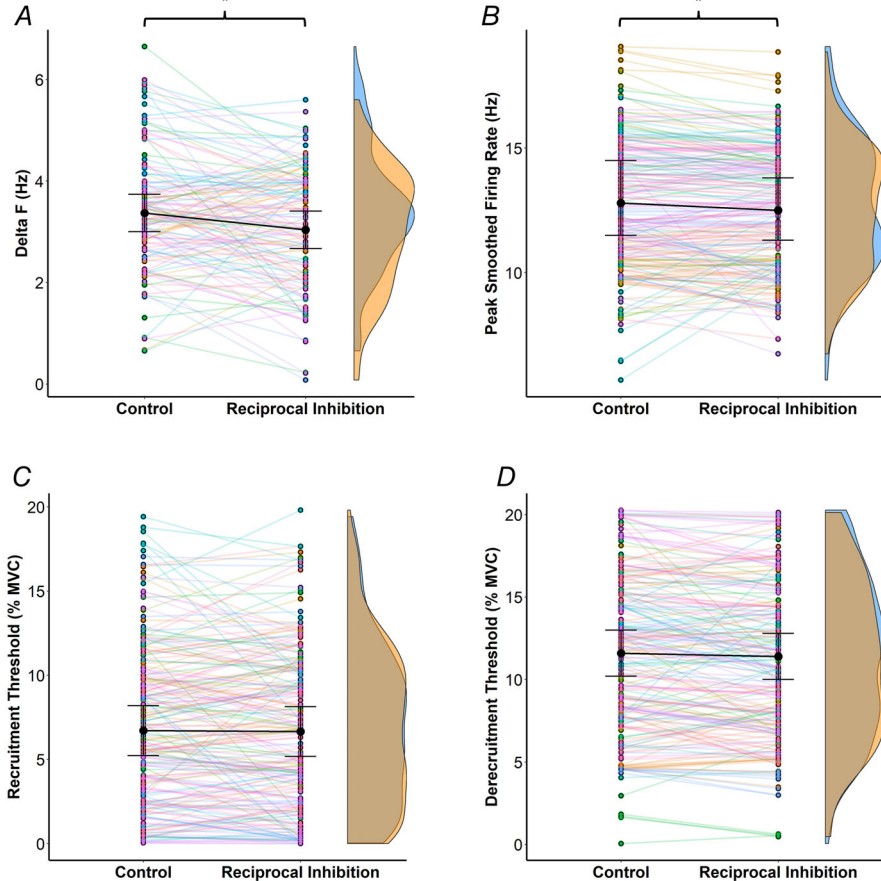

**Figure 2. Data from the paired motor unit technique in control and reciprocal inhibition (common peroneal nerve stimulation) trials**

The different panels illustrate the changes in $\Delta F$ of individual test units (*A*, $n = 130$), peak smoothed firing rate (*B*), recruitment (*C*) and derecruitment thresholds (*D*) of individual motor units ($n = 221$) from control to reciprocal inhibition. Each pair of points represents an individual test unit (*A*) or motor unit (*B*, *C*, *D*), whilst each colour refers to one participant. Estimated marginal means are represented by black circles, with 95% confidence intervals indicated. Kernel density estimation (density curves) of the data is represented on the right by half-violin plots (blue for control and orange for reciprocal inhibition). The peak, valleys, and tails of the density curves can be visually compared to see where control and reciprocal inhibition trials were similar or different. When considering all participants, the repeated measures nested linear mixed-effects models revealed a significant (\**P* < 0.05) decrease in $\Delta F$ and peak smoothed firing rate from control to reciprocal inhibition. Consideration of significant interactions revealed that the decrease in peak smoothed firing rate was only observed in females, and that recruitment and derecruitment thresholds were significantly reduced in males. [Colour figure can be viewed at wileyonlinelibrary.com]

significant effect of RI was observed for any variable (Fig. 5): $T_{vib}$ (5.4 [2.5, 8.2] *vs.* 5.0% MVC [2.5, 7.6]; $P = 0.602$), $EMG_{vib}$ (27.15 [16.27, 38.58] *vs.* 23.43 $\mu$V [17.34, 30.11]; $P = 0.402$), $T_{sust0.5}$ (2.8 [0.6, 4.9] *vs.* 2.5% MVC [1.0, 4.0]; $P = 0.658$), $EMG_{sust0.5}$ (19.26 [12.59, 26.24] *vs.* 18.90 $\mu$V [13.68, 24.39]; $P = 0.878$), $T_{sust3}$ (2.2 [0.3, 4.1] *vs.* 1.6% MVC [0.5, 2.7], $P = 0.072$), $EMG_{sust3}$ (18.61 [12.55, 24.92] *vs.* 16.37 $\mu$V [12.10, 20.81], $P = 0.361$), warm-up (1.3 [−0.3, 2.8] *vs.* 0.9% MVC [−0.1, 2.0], $P = 0.482$). No significant effect of sex ($P = 0.165$–0.988) or sex × intervention interaction ($P = 0.158$–0.960) were observed for any variable. An exploratory analysis with exclusion of participants who did not exhibit an involuntary evoked torque in the time window of interest during the control condition, revealed a significant difference ($P = 0.043$, $n = 7$) in $T_{sust3}$ between control (5.5 ± 5.4% MVC) and RI (3.4 ± 3.3% MVC). Excluding these participants did not change the outcome in the other variables.

**Whole-body relaxation.** One participant was excluded from analysis due to voluntary contractions of the leg during the trials, and another participant withdrew from the study before the completion of this condition ($n = 19$). Significant effects of WBR on $T_{vib}$ [$F_{(1,17)} = 4.93$, $P = 0.040$, $\eta_p^2 = 0.225$] and $T_{sust0.5}$ [$F_{(1,17)} = 4.60$, $P = 0.047$, $\eta_p^2 = 0.213$] were observed (Fig. 6). Both were lower in WBR than control ($T_{vib}$: 4.3 [1.5, 7.3] *vs.* 3.5% MVC [1.1, 5.9], $T_{sust0.5}$: 3.3 [0.8, 5.9] *vs.* 2.1% MVC

[0.3, 4.0]). No significant effects of the intervention were observed for any of the other variables: $EMG_{vib}$ (17.39 [11.05, 24.53] *vs.* 15.32 $\mu$V [10.11, 21.10], $P = 0.052$), $EMG_{sust0.5}$ (15.99 [11.08, 21.58] *vs.* 14.46 $\mu$V [10.52, 18.90], $P = 0.359$), $T_{sust3}$ (2.3 [0.3, 4.5] *vs.* 1.7% MVC [0.2, 3.4], $P = 0.190$), $EMG_{sust3}$ (14.17 [9.94, 18.89] *vs.* 13.66 $\mu$V [10.08, 17.65], $P = 0.734$), warm-up (1.5 [0.3, 2.8] *vs.* 1.0% MVC [−0.1, 2.2], $P = 0.252$). A significant effect of sex was observed for all EMG variables ($EMG_{vib}$: [$F_{(1,17)} = 5.07$, $P = 0.038$, $\eta_p^2 = 0.230$], $EMG_{sust0.5}$: [$F_{(1,17)} = 7.34$, $P = 0.015$, $\eta_p^2 = 0.301$], $EMG_{sust3}$: [$F_{(1,17)} = 7.336$, $P = 0.015$, $\eta_p^2 = 0.301$]), with lower EMG amplitude in females ($EMG_{vib}$: 10.24 [1.91, 18.57] *vs.* 23.15 $\mu$V [14.38, 31.94], $EMG_{sust0.5}$: 9.89 [3.85, 15.93] *vs.* 21.15 $\mu$V [14.79, 27.52], $EMG_{sust3}$: 9.86 [4.63, 15.08] *vs.* 18.43 $\mu$V [12.92, 23.93]). No effect of sex ($P = 0.241$–0.671) was observed for the torque variables, nor a significant sex × intervention interaction ($P = 0.064$–0.862) for any variable. An exploratory analysis with exclusion of participants who did not exhibit an involuntary evoked torque in the time window of interest during control revealed a significant ($P = 0.043$, $n = 10$) decrease in $EMG_{sust0.5}$ from control (18.78 ± 15.13 $\mu$V) to WBR (15.20 ± 11.26 $\mu$V). Outcomes in the other variables were not changed.

## Exploratory analyses

Across individuals there was no significant relationship between averaged control $\Delta F$ scores and averaged control

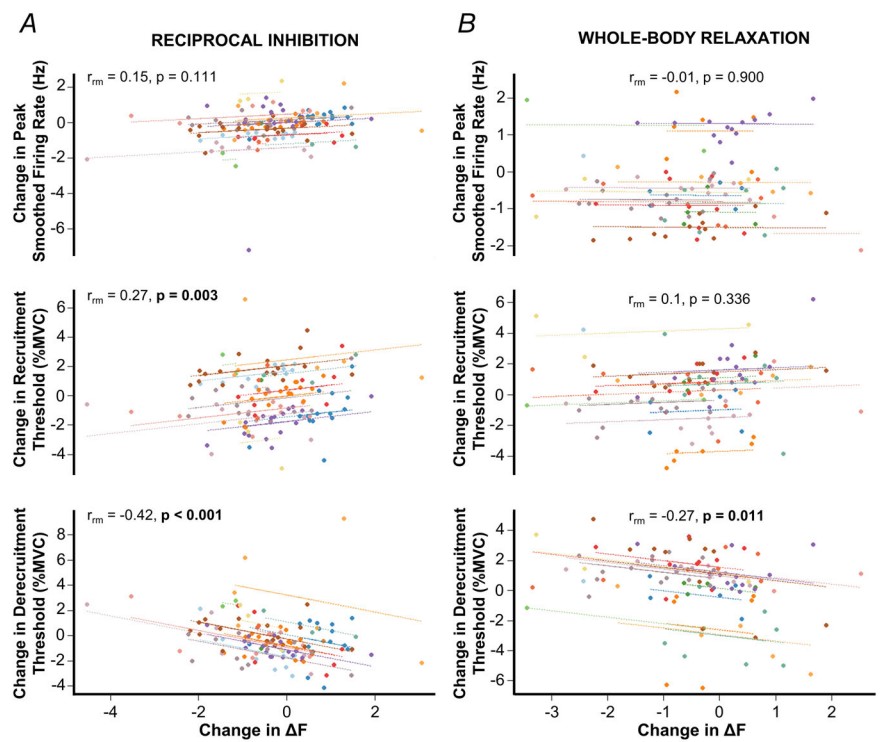

**Figure 3. Association between changes in $\Delta F$ and changes in other dependent variables**
Repeated measures correlations plots illustrating the association between changes in $\Delta F$ and changes in other dependent variables in the reciprocal inhibition (RI) condition (*A*) and in the whole-body relaxation (WBR) condition (*B*). The non-independent nature of the observations is taken into consideration by the repeated measures correlation coefficient ($r_{rm}$), with separate parallel lines fitted to the data from each participant and represented by different colours. In both conditions, a greater decrease in $\Delta F$ of test units was associated with a greater increase in their derecruitment threshold. In RI, but not in WBR, a weak correlation suggests that greater decrease in $\Delta F$ of test units was also associated with a greater decrease in their recruitment threshold. The change in $\Delta F$ and the change in peak smoothed firing rate of test units were not significantly correlated. [Colour figure can be viewed at wileyonlinelibrary.com]

VibStim variables from either condition (Pearson's correlations, all $P > 0.41$; Spearman's correlations, all $P > 0.47$). Moreover, bivariate correlations between average changes in $\Delta F$ per participant and changes in the dependent variables from VibStim between intervention and control showed a significant negative Pearson correlation ($r = -0.69$, $P = 0.009$) with the change in $EMG_{vib}$ in RI. No further significant correlations were found in RI or WBR.

The averaged rectified soleus EMG during repetitive stimulation of the CPN served as a proxy for the magnitude of reciprocal inhibition, and a $19.2 \pm 13.9\%$ decrease in soleus EMG was observed during stimulation. Data were excluded from three participants due to stimulation artefact. Fifteen out of 18 participants exhibited a reduction in soleus EMG which was greater than 14% (range: $-14.0$ to $-54.8\%$), with the remaining three participants exhibiting changes of $-2.1$, $3.7$ and $3.9\%$. The duration of EMG suppression in these 15 participants was $6 \pm 5$ ms.

The magnitude of soleus EMG depression was not correlated with the change in $\Delta F$ or the change in the dependent variables from VibStim in RI. However, a significant negative correlation ($r = -0.610$, $P = 0.007$)

was observed between the pre-stimulus EMG and EMG depression after stimulation.

### Manipulation check and heart rate data

There was a significant increase in self-rated stress in RI (Fig. 7A) during both VibStim [$F_{(1,17)} = 9.03$, $P = 0.008$, $\eta_p^2 = 0.347$] and paired MU trials [$F_{(1,17)} = 5.68$, $P = 0.030$, $\eta_p^2 = 0.262$]. In VibStim, self-rated stress increased from 3.5 [2.9, 4.1] to 4.1 [3.5, 4.8], with no effect of sex ($P = 0.608$) or sex × intervention interaction ($P = 0.522$). In the paired MU technique, self-rated stress increased from 2.9 [2.4, 3.5] to 3.7 [3.0, 4.4], and there was no effect of sex ($P = 0.745$) or an interaction effect ($P = 0.608$).

Conversely, in WBR there was a significant decrease in self-rated stress (Fig. 7B) for VibStim [$F_{(1,17)} = 21.79$, $P < 0.001$, $\eta_p^2 = 0.562$] and paired MU trials [$F_{(1,14)} = 59.50$, $P < 0.001$, $\eta_p^2 = 0.810$]. In VibStim, self-rated stress decreased from 3.3 [2.9, 3.8] to 2.4 [1.9, 3.0], with no effect of sex ($P = 0.291$) or sex × intervention interaction ($P = 0.262$). In the paired MU technique, self-rated stress decreased from 2.5 [2.1, 3.0] to 1.8 [1.4, 2.0], with no effect of sex ($P = 0.745$).

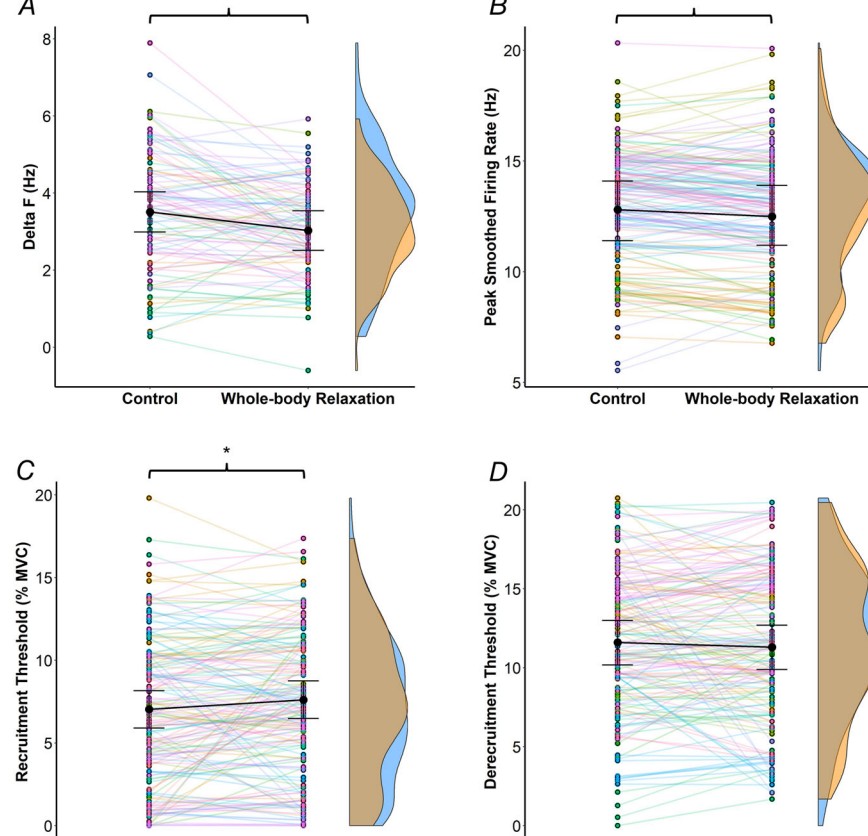

**Figure 4. Data from the paired motor unit technique in control and whole-body relaxation trials**
The different panels illustrate the changes in $\Delta F$ of individual test units (*A*, *n* = 101), peak smoothed firing rate (*B*), recruitment (*C*) and derecruitment thresholds (*D*) of individual motor units (*n* = 174) from control to whole-body relaxation. Each pair of points represents an individual test unit (*A*) or motor unit (*B*, *C*, *D*), whilst each colour refers to one participant. Estimated marginal means are represented in black circles, with 95% confidence intervals indicated. Kernel density estimation (density curves) of the data is represented on the right by half-violin plots (blue for control and orange for whole-body relaxation). The peak, valleys and tails of the density curves can be visually compared to see where control and whole-body relaxation trials were similar or different. The repeated measures nested linear mixed-effects models revealed a significant (**P* < 0.05) decrease in $\Delta F$ and peak smoothed firing rate from control to whole-body relaxation, and an increase in recruitment threshold. [Colour figure can be viewed at wileyonlinelibrary.com]

A significant sex × intervention interaction ($P < 0.001$) revealed that self-rated stress decreased significantly more ($P < 0.001$) in females ($-1.3$ [$-1.6$, $-0.9$], $P < 0.001$) than males ($-0.3$ [$-0.6$, $0.0$], $P = 0.031$).

In RI, there was no significant effect of the intervention on heart rate measured during VibStim (75 beats per minute (bpm) [69, 81] *vs.* 74 [69, 79], $P = 0.374$), no effect of sex ($P = 0.187$) nor an interaction effect ($P = 0.955$). In WBR, a significant interaction between sex and intervention revealed a decrease in heart rate in females (80 bpm [72, 88] *vs.* 77 bpm [69, 85], $P < 0.001$) but not in males (67 bpm [59, 75] *vs.* 66 bpm [58, 74], $P = 0.339$). Heart rate was also significantly higher in females than males in control ($P = 0.029$) but not WBR ($P = 0.70$). No significant correlations were observed between the changes in absolute or relative (% of predicted maximum heart rate [220 bpm – age]) heart rate and changes in the dependent variables in VibStim.

## Discussion

The main findings of the present study were that the contribution of PIC activity to human motoneuron firing in plantar flexor motoneurons ($\Delta F$) was reduced by RI and WBR. Moreover, using VibStim, some variables that quantify the magnitude of involuntary torque and muscle activity produced during and immediately after simultaneous tendon vibration and NMES were found to decrease in WBR but generally remain unaltered in RI. Previous studies show a decrease in PIC activity during inhibitory input using intracellular recordings in animal models, and we now demonstrate this effect with the extraction and analysis of multiple human MUs from

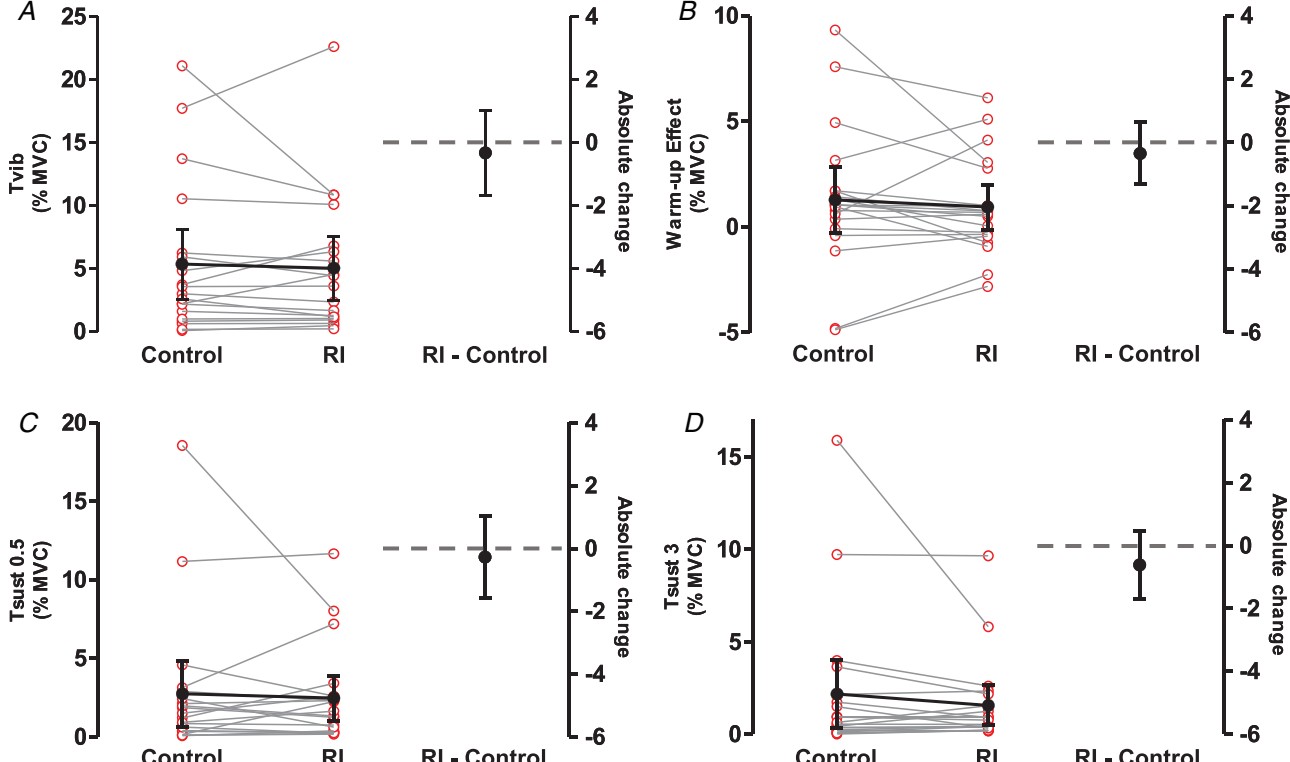

**Figure 5. Changes in the torque-related variables from the control to the reciprocal inhibition (RI) trials during VibStim**

All torques are expressed relative to maximal voluntary contraction torque. On the left side of the graphs, red open circles represent data from individuals with grey lines joining their control and RI values ($n = 20$). Means of control and RI are shown by black filled circles with 95% confidence intervals. Mean change is shown by a black line. The right side of the graphs shows the mean difference between RI and control (black full circle) with 95% confidence intervals. Data on the left side of the graphs are plotted against the left *y*-axis and data on the right against the right *y*-axis, with a grey horizontal dashed line indicating $y = 0$ for the right *y*-axis. $T_{vib}$: torque measured during vibration after the last burst of NMES; Warm-up: difference between $T_{vib}$ and torque after the first burst of NMES; $T_{sust0.5}$ and $T_{sust3}$: torque measured 0.5 and 3 s after vibration cessation, respectively. When considering all participants, there was no evidence of a significant change of any torque-related variables (*A–D*), or any EMG-related variable (data not shown) from control to RI. Nonetheless, an exploratory analysis revealed a significant difference of $T_{sust3}$ between control and RI, when excluding participants who did not exhibit an involuntary evoked torque in the time window of interest during the control condition. [Colour figure can be viewed at wileyonlinelibrary.com]

HD-sEMG signals. Furthermore, the observation for the first time that the contribution of PICs to motoneuron firing is decreased during WBR is important because it demonstrates the potential to manipulate PIC activity through non-pharmacological, neuromodulatory interventions. Of interest, the lack of absolute consistency between the well-established paired MU technique and VibStim in these two interventions could be explained by (1) a floor effect in VibStim resulting from the low magnitude of evoked involuntary torque in most participants combined with the negative effects of the interventions, or (2) the variables quantified in the

current VibStim protocol not being valid estimates of PIC activity.

### Estimated PICs (Δ*F*) are reduced by reciprocal inhibition and whole-body relaxation

The RI-induced decrease in Δ*F* probably resulted from the known powerful effects of inhibitory inputs on motoneuron PIC activity. A reduction in PIC amplitude has been previously observed in intracellular recordings of animal preparations during hyperpolarising currents (Bui et al., 2008; Hounsgaard et al., 1988; Hultborn et al., 2003;

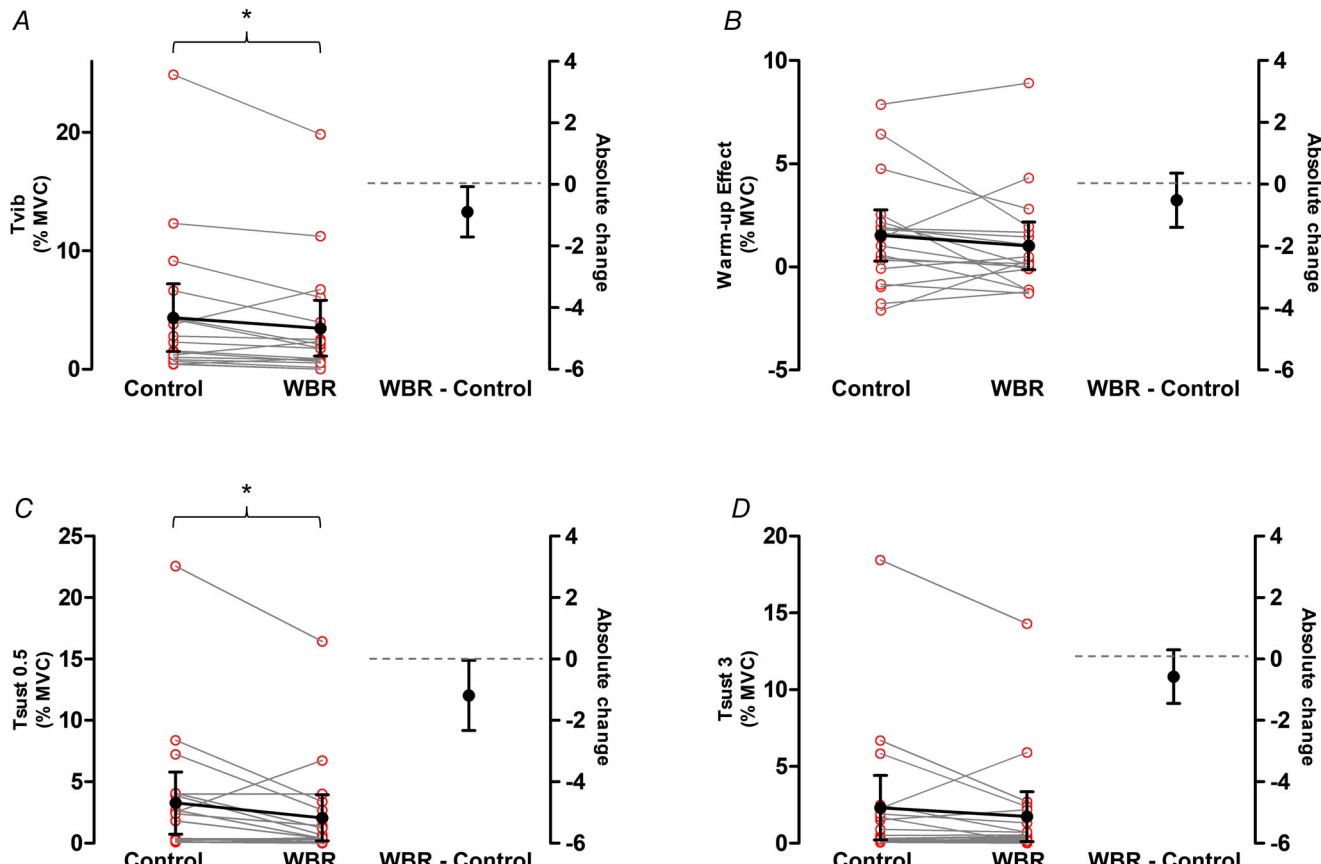

**Figure 6. Changes in the torque-related variables from the control to the whole-body relaxation (WBR) trials during VibStim**

On the left side of the graphs, red open circles represent data from individuals (*n* = 19), normalised to maximal voluntary contraction torque, and grey lines represent individual changes. Means of control and WBR trials are shown in black filled circles, with 95% confidence intervals indicated. Mean change is shown as a black line. The right side of the graphs shows the mean difference between WBR and control (black full circle) with 95% confidence intervals. Data on the left side of the graphs are plotted against the left-hand *y*-axis and data on the right side of the graphs are plotted against the right-hand *y*-axis, with a grey horizontal dashed line indicating *y* = 0 for the right *y*-axis. $T_{vib}$: torque measured during vibration after the last burst of NMES; Warm-up: difference between $T_{vib}$ and torque after the first burst of NMES; $T_{sust0.5}$ and $T_{sust3}$: torque measured 0.5 and 3 s after vibration cessation, respectively. When considering all participants, there was evidence of a significant (*$P < 0.05$) decrease in $T_{vib}$ (*A*) and in $T_{sust0.5}$ (*C*) from control to WBR, and no evidence of a significant change in warm-up (*B*), $T_{sust3}$ (*D*) or in the other EMG-related variables (data not shown). Furthermore, an exploratory analysis revealed a significant decrease in soleus EMG activity measured 0.5 s after vibration cessation from control to WBR, if participants who did not exhibit an involuntary evoked torque in the time window of interest during the control condition were excluded from the analysis (data not shown). [Colour figure can be viewed at wileyonlinelibrary.com]

Kuo et al., 2003) and during lengthening of the antagonist muscle to induce Ia reciprocal inhibition (Hyngstrom et al., 2007). The sensitivity of PICs to inhibition has also been documented in intramuscular recordings of human MUs during cutaneous inhibition (Revill & Fuglevand, 2017). The current study is the first in human MUs to directly suggest that Ia reciprocal inhibition reduces the contribution of PICs to MU firing. This effect is consistent with the preliminary correlational evidence of Vandenberk and Kalmar (2014), showing that a decrease in reciprocal inhibition was associated with an increase in $\Delta F$ in a smaller sample of MUs identified using intramuscular recordings. The 1-Hz electrical stimulation of the antagonist nerve in the present study probably activated Ia inhibitory interneurons, inducing intermittent membrane hyperpolarisation of the motoneurons actively contributing to the contraction, and deactivating or decreasing the activity of the channels underlying motoneuron PICs. An example implication of this is that reciprocal inhibition induced through targeted activation of antagonist Ia afferents (e.g. via electrical stimulation or vibration) might have the therapeutic potential to reduce unwanted PIC-induced activation of motoneurons such as that which underlies the production of involuntary spasms after spinal cord injury (Bennett et al., 2004; DeForest et al., 2020). Moreover, these findings strengthen the hypothesis that this local effect of Ia reciprocal inhibition may be a functionally relevant component to scale the

effects of PICs on motoneuron firing by attenuating the diffuse descending neuromodulation during voluntary contractions (Heckman, Hyngstrom et al., 2008).

To the best of our knowledge, this is the first study to demonstrate a decrease in $\Delta F$ during whole-body relaxation. Given that PIC amplitude is proportional to the level of monoaminergic drive (Heckman, 1994), a decrease in $\Delta F$ might have been caused by an attenuation of serotonergic and noradrenergic release associated with a decrease in whole-body muscular activity (Jacobs et al., 2002), global stress levels and arousal (Valentino & Van Bockstaele, 2008). Potentially, relaxation could be used to attenuate symptoms associated with motoneuron hyperexcitability in people with intact descending mono-aminergic inputs (e.g. spasticity after traumatic brain injury, which appears to be partly induced by an increase of monoaminergic bioavailability (Tsuda et al., 2020)).

Computer simulations (Revill & Fuglevand, 2011) suggest that other phenomena related to motoneuron intrinsic properties such as spike frequency adaptation may also contribute to $\Delta F$. Spike frequency adaptation is a time-dependent reduction in motoneuron firing rate during a constant input (Powers et al., 1999). It is unlikely that PICs contribute to this phenomenon (Zeng et al., 2005) and, speculatively, the contribution of spike frequency adaptation to $\Delta F$ could have been lower if participants had performed triangular ramps rather than trapezoidal ramps (i.e. with a hold-phase). This would

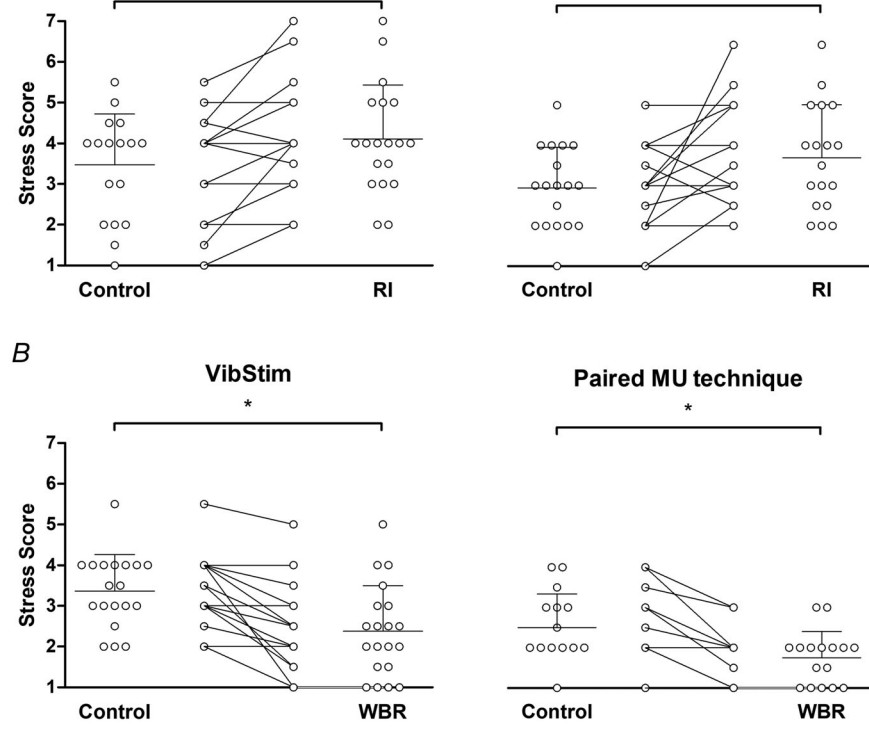

**Figure 7. Self-rated stress scores**
Individual self-rated stress scores reported during control and experimental trials for reciprocal inhibition (*A*) and whole-body relaxation (*B*) conditions. Participants were asked to indicate how relaxed or stressed they felt during each trial. Self-rated stress was significantly (*$P < 0.05$) higher in reciprocal inhibition trials and significantly lower in whole-body relaxation trials. Symbols represent data from individuals. Means are shown in horizontal lines with standard deviations indicated.

have decreased the duration of the contraction and hence the magnitude of spike frequency adaptation, with a consequent increased sensitivity to identify PIC changes during our interventions (Vandenberk & Kalmar, 2014). Thus, the significant decreases in $\Delta F$ values in the present study might have been underestimated.

Importantly, a greater decrease in $\Delta F$ in test units was associated with a greater increase in their derecruitment threshold in both RI and WBR. This suggests that MUs that were derecruited at higher levels of synaptic input (estimated by the firing frequency of the control unit) were also derecruited at higher force levels. Moreover, $\Delta F$ values were not different across recruitment thresholds, suggesting that the magnitude of suprathreshold PICs is similar in motoneurons of different sizes (up to the mid-range of sizes) (Henneman, 1957). This observation is consistent with a recent study of multiple human MUs identified from HD-sEMG signals (Afsharipour et al., 2020) and some animal evidence (Li et al., 2004), but contrary to one study of human MUs identified from intramuscular recordings (Stephenson & Maluf, 2011) and other animal studies (Huh et al., 2017; Lee & Heckman, 1998; Sharples & Miles, 2021); these studies suggest a tendency for stronger PIC magnitudes in higher-threshold motoneurons. As low-threshold motoneurons are more dependent on lasting PICs to maintain firing (Lee & Heckman, 1998), a differential modulation of PIC activity across the motoneuron pool might have been expected during our interventions (Mesquita et al., 2020). Nonetheless, this was not observed in the current study in ramp contractions up to 20% of MVC. Future studies are needed to confirm whether PIC magnitude is invariant with human motoneuron size and if it is differentially modulated by interventions across a wider range of contraction intensities. Finally, our findings suggest that the contribution of PICs to motoneuron firing and its modulation in our interventions were not different between males and females. However, these findings and the differential modulation of other MU firing properties between sexes should be confirmed by future studies with a greater sample size.

### Modulation of firing properties of motor units

The interpretation of changes in MU peak smoothed firing rates, recruitment and derecruitment thresholds is challenging and some methodological constraints should be acknowledged. First, changes in these MU parameters are interdependent, given that participants were asked to follow a force trace whilst generating ramp contractions. Second, decomposition algorithms of HD-sEMG signals allow the identification of a portion of the total population of active MUs and are biased towards MUs with the largest surface action potentials (i.e. larger and more superficial units) (Farina et al., 2010). Finally, all triceps surae

muscles (soleus, GM, and gastrocnemius lateralis) actively contributed to the isometric plantarflexion ramp contractions, but firing events were only identified from GM motoneurons. These synergistic muscles receive minimal common drive during isometric tasks (Hug et al., 2021) and their MUs may be modulated differentially (Hali et al., 2021; Hug et al., 2021). In fact, in the current study, GM EMG amplitudes in the descending phase of the ramp were unexpectedly small and the MUs of some participants had high derecruitment thresholds, which might indicate a greater contribution from synergistic muscles to torque in the descending phase. Thus, we cannot rule out the possibility that some of the observed changes in MU properties were partly influenced by differential effects of our interventions between the disparate components of the triceps surae.

The pattern of modulation of MU parameters was not the same in RI and WBR (see Table 1), although it is not clear why. For example, in RI both recruitment threshold and peak smoothed firing rate tended to decrease, which notionally fits with the recruitment of more MUs to produce the required torque. By contrast, in WBR recruitment thresholds increased while peak smoothed firing rates decreased. In addition, a negative correlation between the recruitment threshold at control and the change in recruitment threshold suggests that the increase in recruitment thresholds was greater in the very early recruited motor units. Possibly, given that PIC channels are activated below the spiking threshold (Afsharipour et al., 2020; Bennett et al., 1998; Bennett, Li, Siu, 2001), lower levels of serotonin and noradrenaline release may have decreased the magnitude of subthreshold activation of PICs. This could have decreased the number of motoneurons near spiking threshold so that a higher net excitatory descending synaptic input was needed to recruit them (Škarabot et al., 2019). This remains speculative, as it is not possible to directly estimate the magnitude of subthreshold PIC activation using the paired MU technique (Afsharipour et al., 2020). Alternatively, or concurrently, WBR led to a greater contribution from synergists early in the contraction.

### Can PIC activity be estimated using VibStim?

VibStim mainly activates Ia afferents from the resting muscle, reflexively recruiting motoneurons. It evokes certain patterns of involuntary torque and muscle activity that resemble PIC-induced behaviour (Mesquita et al., 2021; Trajano et al., 2014). Ideally, the ability to estimate PIC strength both during voluntary contractions with the paired MU technique and at rest with VibStim would make the two techniques complementary. Nonetheless, a decrease was observed only in some VibStim variables ($T_{vib}$ and $T_{sust0.5}$) during WBR and, when including all participants in the analysis, none of the torque or EMG

variables were significantly altered in RI. It might be expected that participants with greater $\Delta F$ scores would also have greater VibStim responses, but no significant correlations to support this were found, nor were changes in VibStim variables positively correlated with changes in $\Delta F$. Thus, we cannot rule out the possibility that the variables quantified in VibStim are not sensitive estimates of PIC activity.

Alternatively, the lack of consistency could be explained by a floor effect in VibStim. Little or no involuntary torque was observed in most of the participants, reducing the likelihood of a significant decrease in the computed variables. Importantly, in previous studies that reported a significant intervention effect (i.e. fatigue (Kirk et al., 2019), electrical parameters (Mesquita et al., 2021), muscle stretching or joint angle change (Trajano et al., 2014)) on VibStim parameters, participants who did not exhibit a progressive increase in force output during a trial were not included in the experiments. An exception is a recent study (Espeit et al., 2021) that used a modified version of the current protocol (with constant rather than intermittent, NMES) and found that electrical stimulation frequency influenced the evoked torque in the participants with higher levels of evoked torque but not in those with lower levels. Similarly, in our current study, reductions in $T_{\text{sust3}}$ and $\text{EMG}_{\text{sust0.5}}$ (in RI and WBR, respectively) were only observed when participants with no involuntary sustained torque were excluded. Moreover, $T_{\text{vib}}$ and $T_{\text{sust0.5}}$ were the only variables to decrease in WBR. They may be less susceptible to a floor effect than other EMG and torque measures because (1) torque data were less variable than EMG data so that small changes might have been more detectable, and (2) torque is typically higher at these times in each trial (see Fig. 1).

A final possibility is that the PIC-like patterns observed in VibStim are not caused by motoneuronal PICs. Modulation of intrinsic muscle properties (Blazevich & Babault, 2019; Frigon et al., 2011) could contribute to the progressive increase of torque during NMES and tendon vibration, although not to EMG activity. Presynaptic post-tetanic potentiation of the Ia input to the motoneurons (Hirst et al., 1981) could contribute to both torque and EMG increases, but neither mechanism can explain the observed self-sustained activity. Nonetheless, according to a recent study in adult cats (Méndez-Fernández et al., 2020), our protocol could have also induced self-sustained firing of spinal cord interneurons, in addition or alternative to alpha motoneurons.

The paired MU technique estimates MU recruitment-derecruitment hysteresis, which has proven to be the most consistent hallmark of PICs, and this method has been validated by animal models (Bennett, Li, Harvey et al., 2001) and computer simulations (Powers & Heckman, 2015). A decrease in $\Delta F$ during RI and WBR in our study further corroborates its sensitivity

to changes in PIC magnitude. Hypothetically, VibStim quantifies PIC-induced self-sustained firing and warm-up (increased firing rates in response to repeated inputs). Given that self-sustained firing and hysteretic firing phenomena probably reflect the same biophysical mechanisms (Binder et al., 2020), changes in $T_{\text{sust0.5}}$, $\text{EMG}_{\text{sust0.5}}$, $T_{\text{sust3}}$, and $\text{EMG}_{\text{sust3}}$ were expected to be associated with changes in $\Delta F$. However, the contribution of PICs to motoneuron firing cannot be estimated in very low-threshold MUs using the paired MU technique as there are no appropriate control MUs. Therefore, different populations of MUs may be assessed by the two techniques, especially if responses to VibStim are small. This means that differential effects across the motoneuron pool need to be considered as an explanation for the lack of change of self-sustained activity; PICs in very low-threshold MUs may have been unaffected by the interventions.

Finally, a limitation of this study is that, although the Achilles tendon vibration was applied by an experienced researcher, pressure on the skin during the trials was not quantified. In extensive piloting and past use of the technique we have not observed a consistent, detectable effect of small pressure variations, although we acknowledge that this may have increased trial-to-trial response variability. Thus, objective measures of skin pressure would improve future studies using VibStim.

## Reciprocal inhibition and whole-body relaxation: methodological and physiological considerations

We are confident that disynaptic reciprocal inhibition was induced in plantar flexor motoneurons with the CPN stimulation protocol as well as significant levels of relaxation in WBR trials. Our supplementary experiment indicated reciprocal inhibition from the ankle dorsiflexors onto the ankle plantar flexors in 16 out of 18 participants. The magnitude of EMG depression in the supplementary experiment and the magnitude of reciprocal inhibitory effect on the estimates of PIC activity were not correlated. However, data were recorded only from soleus in the supplementary experiment and the strength of the inhibitory reflex loop might differ between the three triceps surae components (Henneman et al., 1965). Furthermore, the unexpected significant correlation between the magnitude of soleus EMG depression and the amplitude of baseline EMG suggests that subcutaneous tissue may have acted as a biological low-pass filter, allowing better identification of EMG depression in those participants with higher amplitude of surface EMG. Thus, the EMG depression magnitude might not solely be a function of the effectiveness of inducing reciprocal inhibition in soleus motoneurons. It is important to note that use of a novel decomposition approach allowed MU

firing identification despite the presence of electrical stimulation artefacts in the HD-sEMG signals during RI trials.

A spectrum of arousal states exists when awake, with sleepiness and inattention at one end and stress-induced hypervigilance and panic at the other (Aston-Jones & Bloom, 1981; Ross & Van Bockstaele, 2021). Activity in locus coeruleus (LC) noradrenergic neurons correlates with arousal state and stress, as directly shown in monkeys (e.g. Rajkowski et al., 1997) and indirectly in humans using magnetic resonance imaging (e.g. Naegeli et al., 2018; Sturm et al., 1999). Modulation of diffusely projecting LC neuronal activity evokes diverse noradrenergic responses, including changes in pupil diameter (Joshi et al., 2016) and heart rate (Gurtu et al., 1984; Ter Horst et al., 1991). Importantly, the dendrites and soma of spinal motoneurons also receive LC input (Proudfit & Clark, 1991). In the present study, the parallel decreases in self-rated stress and heart rate, lower $\Delta F$ values, and reduction in two VibStim variables suggest that WBR moderately shifted arousal state, probably decreasing LC activity with a consequent decrease in noradrenergic release onto the motoneurons. Previously, lower peripheral levels of noradrenaline have been shown following passive, seated relaxation (Teixeira et al., 2005), although future studies are needed to better understand other correlates of LC activity during relaxation such as pupil diameter (Joshi et al., 2016) and brain activity using functional magnetic resonance imaging (Turker et al., 2021). Given that spinal motoneurons also receive diffuse serotonergic innervation from the raphe nuclei (Skagerberg & Björklund, 1985), with serotonin release being triggered by motor activity (Noga et al., 2017; Veasey et al., 1995; Wei et al., 2014), it is also possible that a decrease in activity of other, non-tested muscles during WBR led to an additional decrease in serotonin release. Nonetheless, a reduction in monoaminergic release during WBR remains speculative.

Finally, we were interested in whether the magnitudes of RI and WBR effects were similar. Under ideal conditions, inhibitory stimuli to the motoneurons should have greater effect on PIC activity than withdrawal of neuromodulators (Heckman, 1994), given the potential to nearly abolish PICs with the activation of a relatively small proportion of inhibitory synapses (Bui et al., 2006). Moreover, in the paired MU technique, neuromodulator release associated with the voluntary nature of the contractions would be expected (Jacobs et al., 2002; Noga et al., 2017). However, we did not see a striking difference between the two interventions, with RI and WBR decreasing $\Delta F$ by 0.33 Hz [−0.55, −0.12] and 0.49 Hz [−0.72, −0.25], or by ∼10 and ∼14%, respectively. Given the low frequency (1 Hz) used for antagonist nerve stimulation, it is possible that there was opportunity to initiate and increase PIC activity between electrical stimuli. Thus, other protocols with

a higher frequency of inhibition (e.g. high-frequency vibration of the antagonist tendon (DeForest et al., 2020)) might produce greater decreases in PIC activity. A possible attenuation of the inhibitory effects of our RI protocol might also be explained by the unexpected increased levels of self-rated stress during RI, which could have increased noradrenaline release (Valentino & Van Bockstaele, 2008). Moreover, in VibStim, RI might not have been as effective in inducing inhibition in plantar flexor motoneurons due to the constant excitatory, agonist inputs from Ia afferents. These possibilities remain to be explicitly tested in future studies.

## Conclusion

A reduction in the contribution of PICs to motoneuron firing was observed during RI and WBR when examined by changes in human MU firing characteristics. The inhibitory effect of reciprocal inhibition corroborates the importance of focused, local inhibitory circuits in attenuating the diffuse descending neuromodulation and the associated powerful effects of PICs on motoneuron firing during voluntary contractions. Moreover, the attenuation of PICs during RI and WBR demonstrate the potential for electrical stimulation and relaxation techniques to counteract unwanted PIC-induced muscle activation, in conditions marked by motoneuron hyperexcitability. However, since we did not observe absolute consistency between the well-established paired MU technique and the alternative VibStim technique, it remains unclear whether the latter can be used as a proxy for PIC estimation under all physiological conditions.

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

## Additional information

### Data availability statement

Individual data that support the findings of this study are available as a supporting information file. Scripts for semi-automated analysis are available from the corresponding author on request.

### Competing interests

There are no competing interests, financial or otherwise to report regarding this manuscript.

### Author contributions

R.N.O.M., A.J.B., J.L.T. and G.S.T. conceived and designed the work. R.N.O.M. collected the data in the Exercise Physiology Laboratory of Edith Cowan University. R.N.O.M. took the lead in the data analysis. A.H. supervised the decomposition of HD-sEMG signals and J.Š. collaborated. B.A.M. created the scripts and functions to smooth the instantaneous MU firing rate. R.N.O.M. drafted the manuscript. All authors revised the manuscript critically for important intellectual content and approved the final version. All authors agree to be accountable for all aspects of the work in ensuring that questions related to the accuracy or integrity of any part of the work are appropriately investigated and resolved. All persons designated as authors qualify for authorship, and all those who qualify for authorship are listed.

### Funding

The work conducted was supported by a Higher Degree by Research Scholarship awarded to R.N.O.M. J.Š. was supported by Versus Arthritis Foundation Fellowship (ref: 22569). A.H. was supported by the Slovenian Research Agency (project J2-1731 and Programme funding P2-0041).

### Acknowledgements

We thank Tabitha Dearle, Jordan Meester, Caroline Asbury, James O'Loughlin, Cody Wilson, Wayne Poon, Mattias Malchau, Jake Bell and Desire Monty for their help as research assistants during data collection. We thank James Nuzzo for his assistance in the decomposition of EMG signals. We also thank Shih Ching and Sofie Lindeberg for their expert advice on statistical analysis and on developing the manipulation check, respectively.

Open access publishing facilitated by Edith Cowan University, as part of the Wiley – Edith Cowan University agreement via the Council of Australian University Librarians.

### Keywords

bistability, HD-EMG, input-output function, motor neuron, motor neurone

## Supporting information

Additional supporting information can be found online in the Supporting Information section at the end of the HTML view of the article. Supporting information files available:

**Statistical Summary Document**
**Supporting Information for online publication (Data set/Video/Audio only)**
**Peer Review History**

