## [Peer Review History · The Journal of Physiology]

Effects of reciprocal inhibition and whole-body relaxation on persistent inward currents estimated by two different methods

Ricardo N. O. Mesquita, Janet L Taylor, Gabriel S. Trajano, Jakob Škarabot, Ales Holobar, Basílio A. M. Gonçalves, and Anthony Blazevich

DOI: 10.1113/JP282765

Corresponding author(s): Ricardo Mesquita (r.mesquita@ecu.edu.au)

Review Timeline:

Submission Date:	20-Dec-2021
Editorial Decision:	27-Jan-2022
Revision Received:	22-Mar-2022
Editorial Decision:	30-Mar-2022
Revision Received:	04-Apr-2022
Editorial Decision:	06-Apr-2022
Revision Received:	11-Apr-2022
Accepted:	13-Apr-2022

Senior Editor: Richard Carson

Reviewing Editor: Jing-Ning Zhu

Transaction Report:

Dear Mr Mesquita,

Re: JP-RP-2021-282765 "Effects of reciprocal inhibition and whole-body relaxation on persistent inward currents estimated by two different methods" by Ricardo N. O. Mesquita, Janet L Taylor, Gabriel S. Trajano, Jakob Škarabot, Ales Holobar, Basílio A. M. Gonçalves, and Anthony Blazeovich

Thank you for submitting your manuscript to The Journal of Physiology. It has been assessed by a Reviewing Editor and by 2 expert Referees and I am pleased to tell you that it is considered to be acceptable for publication following satisfactory revision.

The reports are copied at the end of this email. Please address all of the points and incorporate all requested revisions, or explain in your Response to Referees why a change has not been made.

NEW POLICY: In order to improve the transparency of its peer review process The Journal of Physiology publishes online as supporting information the peer review history of all articles accepted for publication. Readers will have access to decision letters, including all Editors' comments and referee reports, for each version of the manuscript and any author responses to peer review comments. Referees can decide whether or not they wish to be named on the peer review history document.

Authors are asked to use The Journal's premium BioRender (<https://biorender.com/>) account to create/redrawn their Abstract Figures. Information on how to access The Journal's premium BioRender account is here: <https://physoc.onlinelibrary.wiley.com/journal/14697793/biorender-access> and authors are expected to use this service. This will enable Authors to download high-resolution versions of their figures.

I hope you will find the comments helpful and have no difficulty returning your revisions within 4 weeks.

Your revised manuscript should be submitted online using the links in Author Tasks Link Not Available.

Any image files uploaded with the previous version are retained on the system. Please ensure you replace or remove all files that have been revised.

REVISION CHECKLIST:

- Article file, including any tables and figure legends, must be in an editable format (eg Word)
- Abstract figure file (see above)
- Statistical Summary Document
- Upload each figure as a separate high quality file
- Upload a full Response to Referees, including a response to any Senior and Reviewing Editor Comments;
- Upload a copy of the manuscript with the changes highlighted.

- A potential 'Cover Art' file for consideration as the Issue's cover image;
- Appropriate Supporting Information (Video, audio or data set https://jp.msubmit.net/cgi-bin/main.plex?form_type=display_requirements#supp).

To create your 'Response to Referees' copy all the reports, including any comments from the Senior and Reviewing Editors, into a Word, or similar, file and respond to each point in colour or CAPITALS and upload this when you submit your revision.

I look forward to receiving your revised submission.

If you have any queries please reply to this email and staff will be happy to assist.

Yours sincerely,

REQUIRED ITEMS:

- Author photo and profile. First (or joint first) authors are asked to provide a short biography (no more than 100 words for one author or 150 words in total for joint first authors) and a portrait photograph. These should be uploaded and clearly labelled with the revised version of the manuscript. See Information for Authors for further details.
- You must start the Methods section with a paragraph headed Ethical Approval. If experiments were conducted on humans confirmation that informed consent was obtained, preferably in writing, that the studies conformed to the standards set by the latest revision of the Declaration of Helsinki, and that the procedures were approved by a properly constituted ethics committee, which should be named, must be included in the article file. If the research study was registered (clause 35 of the Declaration of Helsinki) the registration database should be indicated, otherwise the lack of registration should be noted as an exception (e.g. The study conformed to the standards set by the Declaration of Helsinki, except for registration in a database.). For further information see: <https://physoc.onlinelibrary.wiley.com/hub/human-experiments>
- Please upload separate high-quality figure files via the submission form.
- Please ensure that any tables are in Word format and are, wherever possible, embedded in the article file itself.
- A Statistical Summary Document, summarising the statistics presented in the manuscript, is required upon revision. It must be on the Journal's template, which can be downloaded from the link in the Statistical Summary Document section here: https://jp.msubmit.net/cgi-bin/main.plex?form_type=display_requirements#statistics
- Please include an Abstract Figure. The Abstract Figure is a piece of artwork designed to give readers an immediate understanding of the research and should summarise the main conclusions. If possible, the image should be easily 'readable' from left to right or top to bottom. It should show the physiological relevance of the manuscript so readers can assess the importance and content of its findings. Abstract Figures should not merely recapitulate other figures in the manuscript. Please try to keep the diagram as simple as possible and without superfluous information that may distract from the main conclusion(s). Abstract Figures must be provided by authors no later than the revised manuscript stage and should be uploaded as a separate file during online submission labelled as File Type 'Abstract Figure'. Please ensure that you include the figure legend in the main article file. All Abstract Figures should be created using BioRender. Authors should use The Journal's premium BioRender account to export high-resolution images. Details on how to use and access the premium account are included as part of this email.

EDITOR COMMENTS

Reviewing Editor:

The rationale for statistical approaches, the redecomposition of the HD-sEMG signals, the estimation of probability density functions, and the definition of stress score should be described in the Methods, as pointed out by the two reviewers.

Please clarify or improve the statistical methods and experimental design. The sex differences needs to be re-analyzed. The direct relationship between the whole body relaxation and the monoamine level need to be discussed or supported by evidence such as fMRI. The manuscript also needs to be further refined to make it clearer and more concise.

Senior Editor:

There are many positive aspects to this submissions, and the referees and editors share the view that - if appropriately presented, the study has the potential to make an important contribution to the literature. As you will note from the detailed comments and comprehensives reviews, there are however several ways in which the presentation of the study might be enhanced. In the event that you opt to revise the submission, I would ask that all are addressed thoroughly.

REFEREE COMMENTS

Referee #1:

In this manuscript, Mesquita and colleagues show that estimates of PICs in the human plantar flexors can be affected by low frequency stimulation to the common peroneal nerve and by turning down the lights and playing soothing music (what they refer to as whole body relaxation). In general, the paper provides incremental data to support the idea that PICs can be modulated with inhibition or reduced arousal. However, it could benefit from a focus on brevity and consistency in statistical

design and reporting across variables. The methods, though extremely detailed, could be clearer and more concise. Oftentimes, the text is over-complicated and difficult to follow (see below for specifics).

If you are interested in biological sex comparisons, please use female/male, not women/men (gender)

Abstract in general: The hypothesis and importance of inhibition on PICs is unclear from the abstract. Since this is the Journal of Physiology, perhaps the authors could state the reason why this is an advance in our physiological understanding of PICs. In addition, it is misleading when the authors state "To test this, we estimated PICs (n=21, 10 women) 9 using one technique requiring voluntary muscle contractions and another which evokes 10 involuntary contractions" because it suggests that VibStim is a validated method for estimating PICs. Instead, the authors should make it clear that they are both testing the effects of WBR on PICs, and comparing this to RI, which has been shown to modulate PICs, and that they are trying VibStim as a new/alternative approach for estimating PICs. In essence, they are testing whether VibStim is affected similarly to delf. The abstract also states that there were 10 women but PICs were only estimated in 5 and 7 women in the actual experiments. Did 3 of the subjects not produce usable data or were some of the 5 not in the 7 from the other condition? If not, why include 10 here?

Purpose of the investigation: What is the main purpose of this investigation? It is not totally clear what the main motivation of this study was. In one part they are testing two interventions (RI and WBR), in another, they are testing two methods (delf and VibStim), though the main purpose is not entirely clear.

Comparing delf to VibStim: speculation about the reasons why VibStim showed less effective in the examining is provided by the authors, but it seems to be a stretch if the level of PICs estimated in a control condition have no relationship. That is, since there is inter-individual variance in delf, why not try to account for variance in VibStim variables to determine if there is a significant relationship across individuals? For example, the authors state "a comparison of this technique with the well-established paired MU technique has yet to be conducted." If this is a major objective, some sort of relationship between the measures should be examined.

Paired MU analysis: the authors only use test units that were derecruited before the control unit. Did the occurrence of units exhibiting this behavior change in either of the intervention groups? If so, the authors may be critically biasing their estimates of hysteresis. For instance, were there more test units that were derecruited after the control units in control, but not during RI or WBR? What was the incidence of test units that were excluded because of this behaviour?

Did the authors include all MUs or only the units with matched pairs in their analysis of recruitment, derecruitment, and peak discharge characteristics? If only the matched pairs were considered, this would hinder the authors' ability to observe any units that were newly recruited as a result of the intervention. If all units were used, this should be more clearly stated. What percentage of units were identified in control that were not identified in the interventions? For proper sex-comparisons, it would probably be better if all units were compared across the control conditions since paired analysis cannot be used between groups anyways.

RI stimulation condition: Was there an effect of the stimulation (in the RI condition) on the torque traces? For example, was there 1Hz reductions in torque? Please show an example of each condition so that the reader can gain an appreciation of the differences. How well did subjects adjust if there were in fact perturbations in torque? This is quite a low frequency for a 20 second task. Did the derecruitment of MUs correlate with the RI pulses? This may obscure the findings. Were units derecruited and recruited again with some pulses? What was the duration of RI in each pulse? What was the duration of IPSPs? They would likely fluctuate depending on effort.

RI as a form of inhibition: Quantification of RI is mentioned in the methods but not quantified or reported on. Was there a difference between males and females in terms of amplitude or duration of inhibition? Did the amplitude of the RI correlate with the reductions in delf?

Can the authors please provide a more detailed description of the "redcomposition" used? This is not a standard approach, so it would be helpful for readers to understand it fully.

Figure 1B: are the MUs and torque aligned and displayed with the same x-scale? Can the authors please provide an explanation of why many low threshold units that are recruited at <1-2% MVC are derecruited ~3-5 s before the end of the ramp? This is particularly evident in the lowest threshold unit that is recruited just before torque onset, but derecruited 2-3s before returning to rest, at about 7-8% of MVC.

Statistics: Can the authors please provide rationale for varying statistical approaches and software used between the interventions? For example, it is unclear why vibstim stats were not performed in the same way as paired-MU analysis. Since there is multiple trials, this would be a likely candidate for a random effect in a mixed effects model. Again, there were multiple trials in which HR and stress were recorded. Why not take advantage of the variability across trials in a mixed effects model for all comparisons?

Sex comparisons: The sex comparisons are problematic and concerning, especially given the conclusion on lines 24-25. In table 2, it appears that the number of male/female participants is imbalanced (particularly for RI), which could impact the analysis and interpretation of the sex differences. Is it correct that, for RI, only 5 females had available test units, and that delf could only be calculated for 5 females but 10 males? The t-test of the number of motor units between sexes was not

different, but the authors did not run a t-test for the number of test units. In hindsight, a t-test is probably not appropriate for this comparison. Since there were many trials, the proper test would be a mixed effects model to determine if motor unit yield (and test units in a separate comparison) differed between the sexes. Since only matched units were chosen, this could bias the number of units. For example, the males may have more units that were not matched across all trials, but when matched were not different than females. It is necessary that limitations related to the sex-comparisons herein are discussed.

More specific comments:

The human study by Vandenberg & Kalmar does not provide direct evidence that RI affects estimates of PICs. Rather, they explored the relationship between RI (estimated using PSTH and CUSUM) and ePIC, but did not examine the effect of RI on ePIC directly.

P5, Line 6: "net excitatory input" - please consider net synaptic input, which may include an inhibitory component

P5, Line 16-17: With advances in algorithms/software and computing power, the time-consuming aspect is becoming less. One would probably not consider this a major limitation anymore.

P6, Line 26: What does it mean when it says "in RI" - is this referring to the RI condition?

P6, Line 29: Is it the reduced muscle activity and stress or simply reduced "arousal" that causes the effect?

P7, Line 5: Since the data are taken from a larger study, the current manuscript seems a little like patch work. The rationale for such comparisons and the balancing of subjects (or lack thereof) is weak. If there were conditions where the authors show increases in PICs, they would be better packaged with these results to ensure fluidity of the story.

P7, Line 9: Did all subjects complete both studies?

P8, Line 10: What were the other two conditions? On lines 9-11 of P9, this becomes clear, but the order of the conditions is unclear.

P10, Lines 25-26: was 50 or 60 Hz used?

P11, Line 8: why were only 2/3 of the ramps used for analysis? Exactly how were they chosen? Was it completely subjective? Does the same story emerge if all 3 ramps were analysed?

P11, Lines 12-13: How many times did this redecomposition occur?

L11, Lines 26: What is a non-physiological turning point? It would make sense if the firing did not correspond to torque, but if there is a brief increase in torque before a sharp decrease at the end of a ramp, the firing would bend upwards, and this is physiological.

P12, Lines 18-19: Can the authors please provide rationale for averaging values from different behaviours (i.e. ramps)? The ramp behaviour is likely different from trial to trial, which would introduce variability. Why not include this as a factor in the mixed models?

P13, Lines 8-9: During tendon vibration, how was "steady pressure" quantified? If there was no objective measure of pressure, can the authors be certain that the vibration did not impact the sensory feedback of the experienced researcher? How can the authors be sure the pressure was consistent? Was there a loadcell on the tip of the vibrator?

P13, Line 13: Did subjects forcefully hold the straps? Would this increase 'remote' muscle activity and potentially enhance PIC magnitude?

P13: It is unclear why motor units were not also identified during the vibstim. This is not a critical point to address, but it may be of interest to the authors to explore.

P14, Line 4: 8Hz cut-off?

Table 1 is aesthetically pleasing, but would it be better if the authors reported values to give an indication of the magnitudes and which were significant - perhaps using estimated marginal means

P21, Line 8: This comparison is almost certainly underpowered - that is a large difference, a p-value of 0.064 and only 5 females.

What were the estimated marginal mean values for male vs female comparisons (grouped by fixed effect of sex)?

P23, Lines 11-12: How can the authors legitimately conclude that there is not a difference between the number of units, but there are now 67 vs 154 units in females vs males, respectively. That is over 2x the amount of units.

P36: It is difficult to understand the physiological effects of WBR if resting HR did not change. One would expect a reduced heart rate if subjects were more relaxed.

Figure 5A: "RI -" should likely be "RI - Control"

P38, Lines 16-17: If the authors were to correlate ΔF values with measures of vibstim across subjects, they might be able to address this.

P39, Line 4: Channels underlying PICs deactivate slowly, unless there is a powerful inhibition. Without record of the stimulation-induced RI effects, it is unclear whether this stimulation was powerful enough to have affected PICs.

P40, Line 24-26: This claim is vastly underpowered. In addition, Greig Inglis and David Gabriel recently showed that females have higher firing rates than males at low intensities, so it is surprising that the current data shows no sex-related effect at all. In terms of firing rates. Again, it would be easier to understand if the peak firing rates for males and females were reported. In fact, the one place that the rates are reported (Lines 11-12 on P23), female firing rates are about 1 pps higher overall.

P41, Lines 22: the authors state "a reduction in non-linearities of motoneuron firing (i.e., lower levels of acceleration and/or saturation of firing, with more symmetrical firing patterns), which could have increased descending voluntary drive, recruiting and derecruiting motoneurons at lower forces..." - were these non-linearities quantified?

P41, Line 24: "given the compressed thresholds" it is important to acknowledge that this is the compression of recruitment thresholds only of "decomposed" motor units. This does not necessarily mean the thresholds were actually compressed across the entire MU pool.

P42, Line 14: since ΔF is the "standard" - what is the correlation between ΔF and VibStim measures?

P44, Line 29: Why is it 16/18 not 16/21 here?

P45, Line 6-7: Please report the details of these data (i.e., EMG depression due to RI stim). What was the amplitude and duration of suppression?

The effect of inhibition of the PIC is well documented in animal studies. This inhibitory effect is likely to be a fundamental underpinning of normal motor control, as without it, PICs are so powerful they would routinely generate sustained and uncontrolled outputs. So the potential impact is clear.

As for impact, there is only 1 other paper that effectively studies the interaction of inhibition and PICs; this present proposal uses up to date methods and a systematic approach that is uniquely valuable. The results have the potential for numerous citations.

Referee #2:

This is an original research as well as methodological study involving complicated experimental process and large amount of experimental work. The study focuses on comparison of two methods for estimation of PICs in humans. One method is the well-established paired motor unit (MU) technique and the other the ongoing tendon vibration overlaid by short bursts of neuromuscular electrical stimulation (VibStim). The MU method estimates PICs by calculating the delta frequency (ΔF) of MU firing, while the VibStim method estimates PICs by examining the plantar flexor torque. PICs induced by these two methods are estimated under two different conditions, the reciprocal inhibition (RI) and whole-body relaxation (WBR), both assumed to reduce PICs during motoneuron firing in humans.

The experimental results showed that ΔF was significantly decreased by both RI and WBR whereas the torques were reduced by WBR but unchanged in RI. Therefore, the paired motor unit (MU) technique can be used potentially to estimate PICs by non-pharmacological neuromodulatory interventions such as WBR. However, it remains unclear if VibStim can be used as a proxy for PIC estimation.

This study could be helpful for us to understand the mechanisms underlying PIC depression in human motoneurons. Especially, non-pharmacological interventions (electrical stimulation or relaxation) might be used to attenuate unwanted PIC-induced muscle contractions triggered by motoneuron hyperexcitability.

Major concerns:

1. The major concern on this study is the method of whole-body relaxation (WBR) that the authors used as one intervention to reduce monoamine release of the experimental subjects. Although both arousal state (stress, in particular) and the level of voluntary activity influence the noradrenergic and serotonergic systems, the basis for using WBR as a tool to reduce monoamine release and thus PIC strength remains unclear. In sleep state the activity of neurons in both the pontine and medullary groups are shown to be generally unresponsive to a variety of physiological challenges or stressors in animal models. In WBR state, however, the human subjects are maintained in waking state and are allowed to participate in the

designed experiments. Reduction of their monoaminergic system release is unmeasurable during WBR. Therefore the reliability of the experimental results might be uncertain. Although the authors have briefly discussed WBR in Discussion, the physiological basis for WBR is still not clear. It would be good if the authors could clarify this major issue and give a stronger, more detailed physiological background to support their usage of WBR in human study.

2. Following the above discussion, the experimental data showed that all VibStim variables remained unaltered in RI and that in VibStim, the T_{vib} and T_{sus} torque were reduced by WBR but other torque and EMG variables remained unchanged. The inconsistent effect in VibStim suggested that WBR might not be used as a proxy for PIC estimation.

Minor concerns

1. Kernel density estimation (density curves) of the data is represented in Figure 2&4. This process of estimating the probability density function should be briefly described in Methods and the significance of the data should be explained in either text or legends.

2. There is no definition of stress score in the Methods, and this should be described. Furthermore, the rationale of the definition and calculation of the scores for this study should be explained.

3. PICs are mediated by dihydropyridine (DHP) sensitive L-type calcium currents including HVA and LVA or riluzole (or TTX) sensitive slow-inactivated sodium currents. It would be good if the authors could have some discussion about the component of PICs depressed by RI and WBR in this study.

4. Page 38, line 19-20: "Estimates of PICs (ΔF) are reduced by reciprocal inhibition and whole-body" might be stated as "Estimated PICs (ΔF) are reduced by reciprocal inhibition and whole-body relaxation"

This study could be helpful for us to understand the mechanisms underlying PIC depression in human motoneurons. Especially, non-pharmacological interventions (electrical stimulation or relaxation) might be used to attenuate unwanted PIC-induced muscle contractions triggered by motoneuron hyperexcitability.

Dear editors and reviewers,

Thank you for taking the time to review our manuscript and providing us with feedback. We have considered your feedback and made changes accordingly and believe the manuscript has been improved as a result. Please find below a point-by-point explanation of how we have addressed each of your comments, which we hope you will find satisfactory. Changes are highlighted in **blue** (additions) or **red** (deletions). In this document, sometimes deletions were omitted for clarity, but both additions and deletions are highlighted in the revised manuscript. In the responses, the page numbers refer to the manuscript version with highlighted changes.

EDITOR COMMENTS

Reviewing Editor:

The rationale for statistical approaches, the redecomposition of the HD-sEMG signals, the estimation of probability density functions, and the definition of stress score should be described in the Methods, as pointed out by the two reviewers.

Please clarify or improve the statistical methods and experimental design. The sex differences needs to be re-analyzed. The direct relationship between the whole body relaxation and the monoamine level need to be discussed or supported by evidence such as fMRI. The manuscript also needs to be further refined to make it clearer and more concise.

Response

We thank you for the opportunity to revise the manuscript. The rationale for statistical approaches, the redecomposition of the HD-sEMG signals, the estimation of probability density functions and the definition of stress score have been clarified. Further statistical analyses were also conducted, based on Reviewer 1's feedback, and further clarification has been provided. This included the reported sex differences on the number of units. Moreover, speculative interpretations in regard to differential modulation of MU parameters between males and females have been removed from the Discussion. Instead, a note of caution is now explicit in the Discussion, reminding

the reader that there was a low number of participants in our statistical analyses of sex differences. Thus, we cannot draw robust conclusions about the (lack of) sex effects. We have also added a new paragraph in the Discussion to discuss the possible relationship between the whole-body relaxation and the modulation of monoamine level, which includes supporting fMRI evidence and a reminder that a reduction in monoaminergic inputs is speculative. We made a strong effort to reduce the length of the manuscript (see **deletions throughout, in the document with highlighted changes) and improve clarity, although additional information was then asked for by the reviewers. Regardless, the manuscript is slightly shorter overall.**

Senior Editor:

There are many positive aspects to this submissions, and the referees and editors share the view that - if appropriately presented, the study has the potential to make an important contribution to the literature. As you will note from the detailed comments and comprehensives reviews, there are however several ways in which the presentation of the study might be enhanced. In the event that you opt to revise the submission, I would ask that all are addressed thoroughly.

Response

We believe that our study has been improved based on these comprehensive reviews, and we are grateful for the opportunity to revise the manuscript. We have carefully addressed all the comments made by both reviewers.

REFEREE COMMENTS

Referee #1:

In this manuscript, Mesquita and colleagues show that estimates of PICs in the human plantar flexors can be affected by low frequency stimulation to the common peroneal nerve and by turning down the lights and playing soothing music (what they refer to as whole body

relaxation). In general, the paper provides incremental data to support the idea that PICs can be modulated with inhibition or reduced arousal. However, it could benefit from a focus on brevity and consistency in statistical design and reporting across variables. The methods, though extremely detailed, could be clearer and more concise. Oftentimes, the text is over-complicated and difficult to follow (see below for specifics).

Response

We would like to thank the reviewer for taking the time to consider our manuscript. We have conducted additional statistical analyses and clarified the statistical design based on the reviewer's feedback. Changes were also made to the methods to make them clearer and more concise.

If you are interested in biological sex comparisons, please use female/male, not women/men (gender)

Response

We were happy to change the terminology to females/males throughout the manuscript.

Abstract in general: The hypothesis and importance of inhibition on PICs is unclear from the abstract. Since this is the Journal of Physiology, perhaps the authors could state the reason why this is an advance in our physiological understanding of PICs. In addition, it is misleading when the authors state "To test this, we estimated PICs (n=21, 10 women) 9 using one technique requiring voluntary muscle contractions and another which evokes 10 involuntary contractions" because it suggests that VibStim is a validated method for estimating PICs. Instead, the authors should make it clear that they are both testing the effects of WBR on PICs, and comparing this to RI, which has been shown to modulate PICs, and that they are trying VibStim as a new/alternative approach for estimating PICs. In essence, they are testing whether VibStim is affected similarly to delF. The abstract also states that there were 10 women but PICs were only estimated in 5 and 7 women in the

actual experiments. Did 3 of the subjects not produce usable data or were some of the 5 not in the 7 from the other condition? If not, why include 10 here?

Response

We would like to thank the reviewer for raising these important points. We made changes accordingly. First, we now highlight that our findings corroborate the hypothesis that local, focused inhibitory circuits are important to attenuate PIC-induced effects on motoneuron output during voluntary motor. Second, we now explicitly mention that the paired MU technique is well-established for PIC estimation, while VibStim evokes involuntary contractions that could result from PIC activation. Finally, we specify in each result how many participants (and how many female participants) were considered in the analysis. Out of the 21 participants initially recruited, readers can see the reasons why some were excluded in VibStim experiments (initial sentences before presenting VibStim results in the results section), and the breakdown of number of participants with MUs with usable polynomials for further analysis and the number of participants with test units that were included for delta F calculations in Table 2. In the Abstract, further re-wording was done to resolve word count issues. For example, in the last sentence, the statement mentioning that it is unclear whether VibStim can be used as a proxy for PIC estimation was removed, but this information is highlighted in the key points section.

Persistent inward currents (PICs) are crucial for initiation, acceleration, and maintenance of motoneuron firing. As PICs are highly sensitive to synaptic inhibition and facilitated by serotonin and noradrenaline, we hypothesised that both reciprocal inhibition (RI) induced by antagonist nerve stimulation and whole-body relaxation (WBR) would reduce PICs in humans. To test this, we estimated PICs using the well-established paired motor unit (MU) technique. High-density surface electromyograms were recorded from gastrocnemius medialis during voluntary, isometric 20-s ramp, plantarflexor contractions and decomposed into MU discharges to calculate delta frequency (ΔF). Moreover, another technique (VibStim), which evokes involuntary contractions proposed to result from PIC

activation, was used. Plantarflexor torque and soleus activity were recorded during 33-s Achilles tendon vibration and simultaneous 20-Hz bouts of neuromuscular electrical stimulation (NMES) of triceps surae. ΔF was decreased by RI (n=15, 5 females) and WBR (n=15, 7 females). In VibStim, torque during vibration at the end of NMES and sustained post-vibration torque were reduced by WBR (n=19, 10 females), while other variables remained unchanged. All VibStim variables remained unaltered in RI (n=20, 10 females). Analysis of multiple human MUs in this study demonstrates the ability of local, focused inhibition to attenuate the effects of PICs on motoneuron output during voluntary motor control. Moreover, it shows the potential to reduce PICs through non-pharmacological, neuromodulatory interventions such as WBR. The absence of a consistent effect in VibStim might be explained by a floor effect resulting from low-magnitude involuntary torque combined with the negative effects of the interventions.

Purpose of the investigation: What is the main purpose of this investigation? It is not totally clear what the main motivation of this study was. In one part they are testing two interventions (RI and WBR), in another, they are testing two methods (delf and VibStim), though the main purpose is not entirely clear.

Response

We agree with the reviewer that the aim of this study was two-fold. It improves our understanding of mechanisms of PIC depression in human motoneurons and our understanding of strengths and limitations of methodological approaches to (hypothetically) estimate PICs. As a result, it should be of interest to a wide audience (e.g., people interested in the physiological mechanisms of PIC modulation, people interested in modulating PICs in clinical settings, and people interested in the methodological approaches to estimate PICs). Accordingly, in paragraph 2 of the Introduction we state that PICs are modulated by inhibition and neuromodulatory inputs. Then in paragraphs 3 and 4 we introduce the paired MU technique and VibStim. For clarity and based on the reviewer's feedback, we have now added the

following in the beginning of paragraph 5, before stating the aims (Page 6, Lines 19-22):

Besides a better understanding of the capabilities and limitations of methods to non-invasively estimate PICs in humans, investigation of the effects of different interventions with these two methods was expected to provide insight into the effects of PICs on motoneuron firing in different contexts. Thus, the aim of the present study was to examine the effects of both reciprocal inhibition and whole-body relaxation on 1) the contribution of PIC activity to MU firing in plantar flexor motoneurons, estimated using the paired MU technique, and 2) the magnitude of ongoing, involuntary plantar flexion torque and muscle activity assessed during and immediately after simultaneous tendon vibration and NMES application (VibStim), which is assumed to be proportional to PIC activation.

Comparing delF to VibStim: speculation about the reasons why VibStim showed less effective in the examining is provided by the authors, but it seems to be a stretch if the level of PICs estimated in a control condition have no relationship. That is, since there is inter-individual variance in delF, why not try to account for variance in VibStim variables to determine if there is a significant relationship across individuals? For example, the authors state "a comparison of this technique with the well-established paired MU technique has yet to be conducted." If this is a major objective, some sort of relationship between the measures should be examined.

Response

This is a very interesting suggestion. We have conducted additional statistical analysis according to the reviewer's feedback. We have examined both Pearson's and Spearman's rank-order correlations between ΔF scores and VibStim variables, quantified in control trials. This is an exploratory analysis that tries to answer the question "Did those individuals with a greater ΔF scores also exhibit greater VibStim responses?". We did not find any significant correlations and we have now added this

information in the Results (Exploratory Analyses section) and Discussion. You can see below the results of both parametric and non-parametric correlations.

		DeltaF	Tvib	EMGvib	Tsust0.5	EMGsust0.5	Tsust3	EMGsust3	WU
DeltaF	Pearson Correlation	1	.086	-.119	.140	-.132	.155	-.143	.220
	Sig. (2-tailed)		.751	.662	.605	.625	.567	.598	.413

		DeltaF	Tvib	EMGvib	Tsust0.5	EMGsust0.5	Tsust3	EMGsust3	WU	
Spearman's rho	DeltaF	Correlation Coefficient	1.000	.176	-.150	.162	-.179	.097	-.191	.174
		Sig. (2-tailed)	.	.513	.579	.549	.506	.721	.478	.520

Results (Page 35, Lines 10-12):

Across individuals there was no significant relationship between averaged control ΔF scores and averaged control VibStim variables from either condition (Pearson's correlations all $p > 0.41$; Spearman's correlations all $p > 0.47$).

Discussion (Page 43, Lines 13-15):

It might be expected that participants with greater ΔF scores would also have greater VibStim responses, but no significant correlations to support this were found, nor were changes in VibStim variables positively correlated with changes in ΔF. Thus, we cannot rule out the possibility that the variables quantified in VibStim are not sensitive estimates of PIC activity.

Paired MU analysis: the authors only use test units that were derecruited before the control unit. Did the occurrence of units exhibiting this behavior change in either of the intervention groups? If so, the authors may be critically biasing their estimates of hysteresis. For instance, were there more test units that were derecruited after the control units in control, but not during RI or WBR? What was the incidence of test units that were excluded because of this behaviour?

Response

This is an interesting point. However, it is important to note that if the test unit is derecruited after control unit derecruitment, then it would not be possible to compute a ΔF score. The paired MU analysis cannot be used when the control unit is no longer firing. For this study, we are confident that quantifying these occurrences would not improve the manuscript. We are not truly certain about the physiological significance of such a variable (e.g., differential modulation of PIC contribution across the motoneuron pool?), its reliability, or how to quantify it (e.g., incidence of occurrence? Average duration of firing post control unit derecruitment?). Nonetheless, it is a very interesting idea which has not been previously explored in the literature and something we will consider in the future.

Did the authors include all MUs or only the units with matched pairs in their analysis of recruitment, derecruitment, and peak discharge characteristics? If only the matched pairs were considered, this would hinder the authors' ability to observe any units that were newly recruited as a result of the intervention. If all units were used, this should be more clearly stated. What percentage of units were identified in control that were not identified in the interventions? For proper sex-comparisons, it would probably be better if all units were compared across the control conditions since paired analysis cannot be used between groups anyways.

Response

The legend of Table 2 stated that the MUs included in the analysis were those that were tracked from control to experimental trials. We have now clarified that this included all motor units that could be tracked between trials and not only those in appropriate motor unit pairs for delta F calculation (Page 23, Line 3):

*MUs included for analysis: number of MUs with a polynomial fit that could be tracked from control to experimental trials, **regardless of whether they were included in ΔF calculations.** Test units: number of test units used for analysis and*

that could be tracked from control to experimental trials. Pairs: number of pairs that could be tracked from control to experimental trials.

We have now made this more explicit in the Statistical Analysis section of the methods (Page 19, Lines 2-3):

Only MUs that could be tracked between control and experimental trials were included in statistical analyses.

The vast majority of MUs were tracked. Nonetheless, as we mention in the Discussion (Page 41, Lines 13-15), it is important to remember that *decomposition algorithms of HD-sEMG signals allow the identification of a portion of the total population of active MUs and are biased towards MUs with the largest surface action potentials (i.e., larger and more superficial units) (Farina et al., 2010)*. Thus, the known limitations of surface EMG decomposition hinder our ability to examine whether there were newly recruited MUs in our intervention with a sufficiently high level of confidence. Finally, it was not the aim of this study to examine whether there were newly recruited MUs, as we were mainly interested in quantifying changes in the contribution of PICs to motoneuron firing.

The reviewer was wondering what % of MUs was able to be tracked between conditions. This information has now been added in the results section (Page 21, Lines 17-18):

95.8% and 99.0% of MUs were tracked between control and experimental trials in RI and WBR, respectively.

Given the very high proportion of tracked MUs, we don't think that reporting an additional, separate analysis with all MUs for sex-comparisons would improve the manuscript.

RI stimulation condition: Was there an effect of the stimulation (in the RI condition) on the torque traces? For example, was there 1Hz reductions in torque? Please show an example of each condition so that the reader can gain an appreciation of the differences. How well did subjects adjust if there were in fact perturbations in torque? This is quite a low frequency for a 20 second task. Did the derecruitment of MUs correlate with the RI pulses? This may obscure the findings. Were units de-recruited and recruited again with some pulses? What was the duration of RI in each pulse? What was the duration of IPSPs? They would likely fluctuate depending on effort.

Response

The stimulation events were not imported during MU data analysis. Moreover, the 1-Hz stimulation was manually initiated (~10 s before the start of the trial) and thus it is not possible to identify the exact stimulation timings in the ramps. So, we cannot complete a formal analysis to answer the questions asked by the reviewer.

A very small reduction (almost imperceptible) in torque was sometimes observed in the RI ramps, likely induced by the stimulation. However, the magnitude of the drop in force was minimal, not always present throughout the ramp, and only detectible in some participants. Below is an example of 6 out of 12 MUs from one participant in control and RI trials. Although not unequivocal, occasional very small fluctuations in force can be seen in the RI trial that could result from the electrical stimulation. In some other participants these small fluctuations can occasionally be seen with a similar amplitude and not consistently throughout the ramp.

Control

Reciprocal inhibition

Some of the MUs (grey, blue, yellow, and black, in this example) exhibit occasional drops in the average firing frequency in the RI trial. However, this is not seen in all MUs or with a consistent 1-Hz frequency in any participant. A formal analysis to see if these occasional drops in average firing frequency consistently occurred after a stimulus cannot be performed for the reason mentioned above. Moreover, we cannot exclude that the presence of a stimulus artefact may have contributed to these apparent drops in firing rate.

Given the already long paper and our inability to formally quantify these findings we would prefer not to add speculations about this them to the manuscript at this stage. Nonetheless, we have added the following in the Methods (Page 17, Lines 2-4):

The imposition of electrical stimuli did not detectibly affect the participant's ability to follow the on-screen force trace, in the ramp contractions.

RI as a form of inhibition: Quantification of RI is mentioned in the methods but not quantified or reported on. Was there a difference between males and females in terms of amplitude or duration of inhibition? Did the amplitude of the RI correlate with the reductions in delF?

Response

We believe that the reviewer might have missed the following in the “Exploratory Analyses” section in Results (Page 35, Lines 17-27):

*The averaged rectified soleus EMG during repetitive stimulation of the CPN served as a proxy for the magnitude of reciprocal inhibition, and a $19.2 \pm 13.9\%$ decrease in soleus EMG was observed during stimulation. Data were excluded from 3 participants due to stimulation artefact. 15 out of 18 participants exhibited a reduction in soleus EMG which was greater than 14% (range: -14.0 – -54.8%), with the remaining 3 participants exhibiting changes of -2.1, 3.7 and 3.9%. *The duration of EMG suppression in these 15 participants was 6 ± 5 ms.**

The magnitude of soleus EMG depression was not correlated with the change in ΔF or the change in the dependent variables from VibStim in RI. However, a significant negative correlation ($r = -0.610$, $p = 0.007$) was observed between the pre-stimulus EMG and EMG depression after stimulation.

The reviewer also asked whether the magnitude inhibition in our supplementary experiment was different between males and females. We have now conducted an independent t-test to examine this and the reduction in soleus EMG was not different ($p = 0.778$) between females ($-18.2 \pm 8.3 \%$) and males ($-20.4 \pm 19.3 \%$). We believe that adding this information in the manuscript would not be relevant, especially

because there was an unexpected significant correlation between the pre-stimulus EMG and EMG depression after stimulation. As a result, the inter-individual variability observed in the magnitude of inhibition in our supplementary experiment is partly driven by the baseline amplitude of surface EMG (as we mentioned in the discussion: Page 46, Lines 7-12).

Can the authors please provide a more detailed description of the "recomposition" used? This is not a standard approach, so it would be helpful for readers to understand it fully.

Response

A more detailed description has been provided (Page 11, Lines 16-22):

If fewer than 10 MUs were identified after the 2-step automatic tracking, the HD-sEMG signals from one ramp were re-decomposed. In order to fully exploit the frequency bandwidth of the processed HD-sEMG signals, the notch and high-pass differential filters were not applied in the second decomposition run. Newly identified MUs from the second decomposition run were added to those from the first decomposition run. Filters of the new MUs were manually refined in DEMUSE tool and applied to the other trials, one by one, for tracking purposes.

Figure 1B: are the MUs and torque aligned and displayed with the same x-scale? Can the authors please provide an explanation of why many low threshold units that are recruited at <1-2% MVC are derecruited ~3-5 s before the end of the ramp? This is particularly evident in the lowest threshold unit that is recruited just before torque onset, but derecruited 2-3s before returning to rest, at about 7-8% of MVC.

Response

It is indeed an interesting observation, and we already briefly discuss this in our discussion (Page 41, Line 20-24). The highlighted changes below were done to make this idea clearer:

In fact, in the current study, GM EMG amplitudes in the descending phase of the ramp were unexpectedly small and the MUs of some participants had high derecruitment thresholds, which might indicate a greater contribution from synergistic muscles to torque in the descending phase.

We agree that if the modulation of MU firing has a reasonably linear relationship with the change in force, one could wonder whether the EMG and torque data are well synchronised and displayed on the same x-axis scale. We asked ourselves the same question in the early stages of our analysis. As an example, below you can see a “superposition” of all EMG channels, the requested path, and the force trace of one ramp from two participants. Note that the x-axis is represented in number of frames at a sampling frequency of 2000 Hz:

The start of the force does coincide with onset of muscle activity, which suggests that force signals and EMG were well synchronised and displayed on the same x-axis. In the participant on the left, there is very little EMG activity towards the end of the ramp. In the participant on the right, this phenomenon is not as evident.

Below, matching moment to moment modulations also provide evidence that force and EMG signals were well synchronised. The ramp below was not used for analysis but the quick change of force at the end of the ramp (participant quickly relaxed and contracted the muscle) occurs with an associated modulation of MU firing.

We thus concluded that low-EMG amplitudes with relatively high derecruitment thresholds (as pointed out by the reviewer) in some participants were likely a consequence of a greater contribution of synergistic muscles towards the end of the ramp. This phenomenon has been recently documented in plantar flexors and might occur because the three muscles composing triceps surae share minimal common drive (Hug et al., 2021).

Statistics: Can the authors please provide rationale for varying statistical approaches and software used between the interventions? For example, it is unclear why vibstim stats were not performed in the same way as paired-MU analysis. Since there is multiple trials, this would be a likely candidate for a random effect in a mixed effects model. Again, there were multiple trials in which HR and stress were recorded. Why not take advantage of the variability across trials in a mixed effects model for all comparisons?

Response

We would like to clarify that different statistical approaches and software were used for the results emanating from the different techniques, but statistical approaches

were consistent between interventions, as described in the statistical analysis section. Different software were used for different statistical tests and this was simply due to the personal preference of the first author who conducted the statistical analysis. We would also like to clarify that HR was not measured during ramp contractions, as mentioned in Page 18, Lines 13-15:

ECG data were not recorded during the paired MU technique trials, as pilot testing demonstrated that ECG electrode connection increased the background noise in HD-SEMG signals.

The main reason to use repeated measures nested linear mixed-effects models to examine the effect of each intervention on MU variables was the nested data structure in each trial (i.e., each participant had a different number of observations [motor units] and the number of observations was different between participants). We acknowledge that data from both VibStim trials could have also been used in a mixed-effects model. Nonetheless, the data within each VibStim trial was not nested (i.e., only one value per participant per trial), and thus the analysis of the dependent variables with a mixed ANOVA is appropriate and sufficiently robust. All ANOVA assumptions were carefully examined.

Sex comparisons: The sex comparisons are problematic and concerning, especially given the conclusion on lines 24-25. In table 2, it appears that the number of male/female participants is imbalanced (particularly for RI), which could impact the analysis and interpretation of the sex differences. Is it correct that, for RI, only 5 females had available test units, and that delF could only be calculated for 5 females but 10 males? The t-test of the number of motor units between sexes was not different, but the authors did not run a t-test for the number of test units. In hindsight, a t-test is probably not appropriate for this comparison. Since there were many trials, the proper test would be a mixed effects model to determine if motor unit yield (and test units in a separate comparison) differed between the sexes. Since only matched units were chosen, this could bias the number of units. For example, the males may have

more units that were not matched across all trials, but when matched were not different than females. It is necessary that limitation related to the sex-comparisons herein are discussed.

Response

We made several changes related to the sex comparisons, which have improved the manuscript. Rather than examining differences in the number of decomposed MUs between males and females, we have now tested for differences in the number of MUs that were used in further statistical analyses (i.e., excluding the small portion of MUs that were initially identified but excluded due to poor polynomial fitting or because they were not tracked between conditions) as well as in the number of test units (as suggested by the reviewer). Importantly, the structure of the data related to the number of units is not nested/hierarchical. This happens because average scores were used in the statistical analysis of secondary MU parameters (as explained in a response to a comment below), and pairs from two trials were merged to generate a single ΔF score per test unit (as explained in detail in our methods). It is then suitable to conduct separate independent t-tests to examine whether there were differences in the number of units between males and females. Our new statistical analyses did not reveal significant differences, as it is mentioned in the results section (Pages 21-22, Lines 20-4):

For participants with MUs included in statistical analysis, there was no significant difference in the number of MUs per participant between females and males in RI ($p = 0.223$; females: 9.6 ± 7.0 , males: 14.0 ± 7.3) or WBR ($p = 0.221$; females: 9.0 ± 3.8 , males: 12.3 ± 6.0). There was also no difference in the number of test units in RI ($p = 0.350$; females: 6.6 ± 6.1 , males: 9.7 ± 5.7) or WBR ($p = 0.152$; females: 5.0 ± 3.4 , males: 8.3 ± 4.7). ~~There was no evidence of a statistically significant difference in the number of identified MUs between men and women in RI ($p = 0.064$, $d = 0.860$, $CL = 73\%$, women: 8.0 ± 6.9 , men 14.1 ± 7.4) or WBR ($p = 0.111$, $d = 0.748$, $CL = 70\%$, women: 7.5 ± 5.3 , men: 11.8 ± 6.2). The lack of a significant difference in the number of MUs between groups remained when including both conditions in the same~~

~~statistical test and considering the average ($p = 0.067$) or maximum ($p = 0.081$) number of MUs of each participant across conditions.~~

Although significant differences were not observed, we do agree with the reviewer that strong conclusions related to this exploratory examination should be avoided. Thus, in the Discussion we now advocate caution regarding findings related to (the lack of) effects of sex and have eliminated speculation related to differential modulation of MU parameters between males and females.

(Pages 40-41, Lines 30-5):

Finally, our findings suggest that the contribution of PICs to motoneuron firing and its modulation in our interventions were not different between males and females. However, these findings and the differential modulation of MU firing properties between sexes should be confirmed by future studies with a greater sample size. To the best of our knowledge, this is the first study to examine an effect of sex on PICs.

(Pages 41-42, Lines 27-9):

~~The differential effect of RI on MU behaviour between women and men might be partly explained by a more compressed range of recruitment thresholds (~0-12 vs. ~0-19% MVC) in women. In men, while MUs achieved a similar peak smoothed firing rate, the inhibitory effect of the protocol likely led to a reduction in non-linearities of motoneuron firing (i.e., lower levels of acceleration and/or saturation of firing, with more symmetrical firing patterns), which could have increased descending voluntary drive, recruiting and derecruiting motoneurons at lower forces. Given the compressed thresholds in women, the same strategy could not be used to the same extent and/or was more difficult to identify in our statistical tests. Rather, a reduction in non-linear behaviour of motoneurons could have led to~~

~~*slower peak smoothed firing rates while preserving a similar average firing rate or recruiting additional MUs not identified through our methods.*~~

Finally, and as mentioned in the response to a previous comment, a very high percentage of units in our study were tracked between control and intervention trials. Thus, additional analysis including non-tracked MUs is not warranted.

More specific comments:

The human study by Vandenberg & Kalmar does not provide direct evidence that RI affects estimates of PICs. Rather, they explored the relationship between RI (estimated using PSTH and CUSUM) and ePIC, but did not examine the effect of RI on ePIC directly.

Response

We would like to thank the reviewer for pointing this out. We agree that the study by Vandenberg & Kalmar (2014) did not directly examine the effect of RI on PICs of human spinal motoneurons and this could be clearer in our manuscript. Evidence by Vandenberg & Kalmar (2014) is correlational, given that a decrease in estimated RI was associated with an increase in ΔF , across different joint angles. We believe that the changes below clarify this, emphasising the novelty of our findings.

Introduction, Page 4, Lines 20-24:

*In fact, PIC activity can be markedly reduced by inhibitory ionotropic synaptic input such as disynaptic Ia reciprocal inhibition (RI), **with robust evidence in animal models (Kuo et al., 2003; Hyngstrom et al., 2007) and preliminary observations in humans (Trajano et al., 2014; Vandenberg & Kalmar, 2014).***

Introduction, Page 6, Lines 7-10:

Additionally, the magnitude of involuntary force and muscle activity is muscle length dependent (Trajano et al., 2014), consistent with the effect of reciprocal inhibition observed in vivo using voltage clamp (Hyngstrom et al., 2007) ~~and paired MU techniques (Vandenberk & Kalmar, 2014).~~

Discussion, Page 38, Lines 6-10:

Previous studies show a decrease in PIC activity during inhibitory input using intracellular recordings in animal models ~~or in a relatively small number of human MUs,~~ and we now demonstrate this effect with the extraction and analysis of multiple human MUs from HD-sEMG signals.

Discussion, Page 38-39, Lines 28-6:

The sensitivity of PICs to inhibition has also been documented in intramuscular recordings of human MUs during ~~Ia reciprocal inhibition (Vandenberk & Kalmar, 2014) and~~ cutaneous inhibition (Revill & Fuglevand, 2017). The current study is the first in human MUs to directly suggest that Ia reciprocal inhibition reduces the contribution of PICs to MU firing. This effect is consistent with the preliminary correlational evidence of Vandenberk & Kalmar (2014), showing that a decrease in reciprocal inhibition was associated with an increase in ΔF in a smaller sample of MUs identified using intramuscular recordings.

P5, Line 6: "net excitatory input" - please consider net synaptic input, which may include an inhibitory component

Response

We agree with the reviewer's suggestion. This has been changed throughout the manuscript.

P5, Line 16-17: With advances in algorithms/software and computing power, the time-consuming aspect is becoming less. One would probably not consider this a major limitation anymore.

Response

While we agree that the time-consuming aspect is becoming less, we intended to tell the reader that EMG decomposition and MU analysis are not a "black box", requiring procedures which are more time-consuming and complex than other types of analysis (e.g., quantification of VibStim variables). We did some re-wording, using the adverb "*relatively*", to make this idea clearer (Page 5, Lines 14-17):

*These approaches exploit the 1-to-1 ratio between the firing rate of a motoneuron and MU action potentials in the muscle, providing a unique window into the central nervous system, but require **computational procedures that are relatively time-consuming and complex.***

P6, Line 26: What does it mean when it says "in RI" - is this referring to the RI condition?

Response

That is correct. We understand that the use of this acronym could have been confusing. Thus, RI and WBR are no longer defined in the Introduction. They are defined in the Methods section instead when describing the conditions. Now, throughout the manuscript we use *reciprocal inhibition* when we refer to the physiological mechanism and *whole-body relaxation* when we refer to the state of being relaxed, and *RI* and *WBR* when we refer to the conditions in our study. We would like to thank the reviewer for pointing this out.

P6, Line 29: Is it the reduced muscle activity and stress or simply reduced "arousal" that causes the effect?

Response

That is a good question and we do think that a reduction of arousal could also contribute, as previously mentioned in the Introduction. The idea of a reduction in arousal has been added to this sentence (Pages 6-7, Lines 29-4):

*It was hypothesised that PIC strength estimated by both techniques would decrease both **with reciprocal inhibition**, given that PICs are strongly reduced by inhibitory inputs, and in **whole-body relaxation**, speculatively as a result of a decreased serotonergic and noradrenergic release onto motoneurons associated with the reduced muscle activity, global stress levels, **and arousal**.*

P7, Line 5: Since the data are taken from a larger study, the current manuscript seems a little like patch work. The rationale for such comparisons and the balancing of subjects (or lack thereof) is weak. If there were conditions where the authors show increases PICs, they would be better packaged with these results to ensure fluidity of the story.

Response

Given the complexity and extension of experiments, including only the conditions in which a reduction in PIC activity were hypothesised improves the readability of the manuscript. It also allows more space to discuss the physiological insights about PIC depression in human motoneurons, which is highly relevant for the Journal of Physiology readership. We already have 17 pages of Results and 12 pages of Methods (including figures). Including the conditions in which we hypothesised increases in PIC activity would double the length of the results and result in ~2 additional pages of methods. Moreover, both the Reviewing Editor and Reviewer 1 have expressed that the paper could benefit from being more concise and including two additional conditions would have the opposite effect.

Nonetheless, if the editors think it is appropriate, we would be happy to submit both manuscripts simultaneously, as companion papers.

P7, Line 9: Did all subjects complete both studies?

Response

We think that the reviewer is asking whether all participants completed all experimental interventions. All 21 participants (with the exception of one participant, as mentioned in the results) completed all interventions. However, as described, not all participants provided usable motor units and some technical problems reduced n for some VibStim variables.

P8, Line 10: What were the other two conditions? On lines 9-11 of P9, this becomes clear, but the order of the conditions is unclear.

Response

Page 8 (Lines 15-17) stated that the order of the conditions was randomised. We have adjusted the wording to make this clearer:

On experimental days, VibStim and paired MU techniques were performed under four experimental conditions. ~~Two experimental conditions took place on each day in a randomised order.~~ The four conditions were performed in a randomised order over two experimental days with two conditions per day.

P10, Lines 25-26: was 50 or 60 Hz used?

Response

The notch filter in the software tool used (DEMUSE tool) adapts the notch filter automatically to 50 Hz or 60 Hz line interference and its higher harmonics. We have changed the text to “50 Hz” to match the line interference in Australia.

P11, Line 8: why were only 2/3 of the ramps used for analysis? Exactly how were they chosen? Was it completely subjective? Does the same story emerge if all 3 ramps were analysed?

Response

The ability to successfully follow a force trace during ramp contractions requires familiarisation and varies between individuals and between trials. By choosing 2 out of 3, we were able to include the two best attempts and improve the quality of the data. The two ramps in each block were manually selected based on criteria that are already described in Page 11 (Lines 14-16). These were determined by visual inspection, and this detail was added to the sentence:

The ramps for further analysis were selected by visual inspection. Selection was based on the number of MUs identified, smoothness and adherence to the torque template, and MU firing profiles.

Moreover, it is important to note that high levels of reliability for ΔF and Vibstim variables have been previously reported between consecutive trials (Trajano et al., 2014, 2020).

P11, Lines 12-13: How many times did this redecomposition occur?

Response

Redecomposition of the EMG signals was conducted if less than 10 reliable MUs were identified after a quick manual editing of the spike trains. This process was conducted in 10 participants in RI and in 12 participants in WBR. The redecomposition process itself was conducted once per participant, which included 50 decomposition runs.

L11, Lines 26: What is a non-physiological turning point? It would make sense if the firing did not correspond to torque, but if there is a brief increase in torque before a sharp decrease at the end of a ramp, the firing would bend upwards, and this is physiological.

Response

Thank you for pointing this out. We agree with the reviewer that the expression “non-physiological turning point” could be misleading. We made some changes accordingly (Page 12, Lines 8-13):

*All polynomials were visually inspected and if **edge effects were observed at MU recruitment or derecruitment with a 5th-order polynomial (i.e., a clear mismatch between the change in the smoothed and instantaneous firing rate)**, then a 4th-order polynomial was used. If **edge effects were observed for both 5th- and 4th-order polynomials**, the MU from that specific trial was not included in further analyses.*

We would like to point out that this is the first PIC study, to our knowledge, in which a visual inspection of the smoothed firing rates was conducted. Including polynomials with edge effects could have affected correlation coefficients between MU firings and could have affected ΔF values if these edge effects were observed in the control unit. Therefore, we believe this is a strength of our study.

P12, Lines 18-19: Can the authors please provide rationale for averaging values from different behaviours (i.e. ramps)? The ramp behaviour is likely different from trial to trial, which would introduce variability. Why not include this as a factor in the mixed models?

Response

Generally speaking, there is a true value that we try to approximate when we measure a certain variable. We acknowledge that different approaches of data management before statistical analysis can be used to achieve this goal. In this case, given that only 2 trials per participant were measured, we felt that it was a better compromise to generate averages of our secondary variables in the paired MU technique trials (i.e., recruitment threshold, derecruitment threshold and peak smoothed firing rate) to reduce the variability of the measure. Given the low number of trials, the mixed model may not be able to achieve robust estimates of variance, if the number of trials was included as a factor. Perhaps more importantly, our main variable (ΔF) was not

subject to a simple arithmetic mean between the two trials. Rather, a novel two-ramp, multi-control, repeated measures approach was used to compute the ΔF scores that were then used in the statistical analysis (merged pairs from both trials were used, as described in detail in Pages 12-13, Lines 28-11).

P13, Lines 8-9: During tendon vibration, how was "steady pressure" quantified? If there was no objective measure of pressure, can the authors be certain that the vibration did not impact the sensory feedback of the experienced researcher? How can the authors be sure the pressure was consistent? Was there a loadcell on the tip of the vibrator?

Response

There was no loadcell on the tip of the vibrator, so we could not quantify the pressure of the vibrator on the skin. The following paragraph has now been added to the end of the section "Can PIC activity be estimated using VibStim?" (Page 45, Lines 19-24):

Finally, a limitation of this study is that, although the Achilles tendon vibration was applied by an experienced researcher, pressure on the skin during the trials was not quantified. In extensive piloting and past use of the technique we have not observed a consistent, detectable effect of small pressure variations, although we acknowledge that this may have increased trial-to-trial response variability. Thus, objective measures of skin pressure would improve future studies using VibStim.

P13, Line 13: Did subjects forcefully hold the straps? Would this increase 'remote' muscle activity and potentially enhance PIC magnitude?

Response

During the trials, participants were asked to hold the shoulder straps of the chair, but not forcefully. Our pilot testing revealed that asking participants to hold the straps provided greater standardisation between trials vs. not holding the straps. Not holding the straps often resulted in varying magnitude of muscle activity of the upper

arms, which could variably influence PIC magnitude due to “remote” muscle activity, as suggested by the reviewer.

P13: It is unclear why motor units were not also identified during the vibstim. This is not a critical point to address, but it may be of interest to the authors to explore.

Response

We agree with the reviewer that future studies could usefully explore this possibility. It would come with certain methodological challenges, such as possible difficulties in decomposing EMG with NMES and vibration artefacts, and the fact that the stimulating electrodes already cover a big portion of the gastrocnemii. Nonetheless, this is something we would like to explore in the future.

P14, Line 4: 8Hz cut-off?

Response

Thank you. This was a typo and it has been amended.

Table 1 is aesthetically pleasing, but would it be better if the authors reported values to give an indication of the magnitudes and which were significant - perhaps using estimated marginal means

Response

According to the journals' guidelines, *the same information should not be presented in both tabular and graphical forms.* We already present estimated marginal means in other figures. We believe that the simple and aesthetically pleasing characteristics of this table will help the reader to quickly understand what the main results of the present study were.

P21, Line 8: This comparison is almost certainly underpowered - that is a large difference, a p-value of 0.064 and only 5 females.

Response

As mentioned in a previous comment, the outputs of this specific analysis have been eliminated from our manuscript. Moreover, a note of caution is now explicit in the Discussion regarding the examination of effects of sex in our study.

What were the estimated marginal mean values for male vs female comparisons (grouped by fixed effect of sex)?

Response

We are not sure to which comparison the reviewer is referring. Perhaps to the comparison from the previous comment? This analysis is no longer in the manuscript, and it was replaced by other exploratory analysis (independent t-tests) to examine differences in the number of units between males and females.

P23, Lines 11-12: How can the authors legitimately conclude that there is not a difference between the number of units, but there are now 67 vs 154 units in females vs males, respectively. That is over 2x the amount of units.

Response

We used an independent t-test to test whether there was statistical evidence that the associated means of the number of units in males and females who were included in statistical analyses were significantly different. There was no evidence that they were significantly different. Equal variances were assumed (Levene's test). The generated *t* value of this test does not simply take into consideration the sum of all values from each group (i.e., 67 vs 154); that would have been especially problematic here given the different number of male and female participants. The final *t* value (and associated p-value) also depends on the means of each group, sample size of each group, standard

deviation of each group, and pooled standard deviation. We did not have evidence to reject the null hypothesis, but we did add a note of caution in the Discussion that future studies should confirm our results related to the (lack of) effect of sex, due to the low number of participants in each group.

P36: It is difficult to understand the physiological effects of WBR if resting HR did not change. One would expect a reduced heart rate if subjects were more relaxed.

Response

Interestingly, as we mention in Results, there was a significant decrease in HR in females ($p < 0.001$) but not in males ($p = 0.339$). HR in females was also significantly higher than males. Accordingly, the decrease in self-rated stress was also greater in females. It is possible that the voluntary or evoked contractions impair the ability to observe a relaxation-induced decrease in HR in all participants. Importantly, self-rated stress was markedly lower in WBR trials, which gives us confidence that we were successful in inducing relaxation. Below you can see individual data of changes in HR.

Figure 5A: "RI -" should likely be "RI - Control"

Response

In the submitted manuscript it says "RI - Control" in the figure. We wonder if there was some visualisation issue with the figure when the reviewer viewed the document? We have now uploaded a separate high-quality file for each figure.

P38, Lines 16-17: If the authors were to correlate ΔF values with measures of vibstim across subjects, they might be able to address this.

Response

This has been addressed in a previous comment.

P39, Line 4: Channels underlying PICs deactivate slowly, unless there is a powerful inhibition. Without record of the stimulation-induced RI effects, it is unclear whether this stimulation was powerful enough to have affected PICs.

Response

An evident EMG suppression immediately after each stimulus (supplementary experiment), lower ΔF scores (main experiment), and the known powerful effects of brief inhibitory inputs on PIC activity (previous studies) allow us to suggest that: (1) disynaptic reciprocal inhibition was induced in plantar flexor motoneurons with the common peroneal nerve stimulation, (2) there was a lower contribution of PICs to motoneuron firing, and (3) the former likely caused the latter.

P40, Line 24-26: This claim is vastly underpowered. In addition, Greig Inglis and David Gabriel recently showed that females have higher firing rates than males at low intensities, so it is surprising that the current data shows no sex-related effect at all. In terms of firing rates. Again, it would be easier to understand if the peak firing rates for males and females

were reported. In fact, the one place that the rates are reported (Lines 11-12 on P23), female firing rates are about 1 pps higher overall.

Response

This claim has been eliminated, as mentioned in a previous comment.

P41, Lines 22: the authors state "a reduction in non-linearities of motoneuron firing (i.e., lower levels of acceleration and/or saturation of firing, with more symmetrical firing patterns), which could have increased descending voluntary drive, recruiting and derecruiting motoneurons at lower forces..." - were these non-linearities quantified?

Response

This interpretation has been eliminated, as mentioned in a previous comment.

P41, Line 24: "given the compressed thresholds" it is important to acknowledge that this is the compression of recruitment thresholds only of "decomposed" motor units. This does not necessarily mean the thresholds were actually compressed across the entire MU pool.

Response

This interpretation has been eliminated, as mentioned in a previous comment.

P42, Line 14: since delta F is the "standard" - what is the correlation between delF and VibStim measures?

Response

This has been addressed in a previous comment.

P44, Line 29: Why is it 16/18 not 16/21 here?

Response

As mentioned in a previous response, the reviewer might have missed some information from the section “Exploratory analyses”, in the Results. It is stated that *data were excluded from 3 participants due to stimulation artefact.*

P45, Line 6-7: Please report the details of these data (i.e., EMG depression due to RI stim). What was the amplitude and duration of suppression?

Response

The amplitude of suppression is already reported in the section “Exploratory analyses”, in the results. We have now added the average duration of suppression (Page 35, Lines 22-23): *The duration of EMG suppression in these 15 participants was 6 ± 5 ms.*

The effect of inhibition of the PIC is well documented in animal studies. This inhibitory effect is likely to be a fundamental underpinning of normal motor control, as without it, PICs are so powerful they would routinely generate sustained and uncontrolled outputs. So the potential impact is clear.

As for impact, there is only 1 other paper that effectively studies the interaction of inhibition and PICs; this present proposal uses up to date methods and a systematic approach that is uniquely valuable. The results have the potential for numerous citations.

Response

We would like to thank the reviewer for this positive feedback. We agree that the results of this study provide novel physiological insights, suggesting important implications in motor control.

References

Farina D, Holobar A, Merletti R & Enoka RM (2010). Clinical Neurophysiology Decoding the neural drive to muscles from the surface electromyogram. Clin Neurophysiol 121, 1616–1623.

Hug F, Del Vecchio A, Avrillon S, Farina D & Tucker K (2021). Muscles from the same muscle group do not necessarily share common drive: evidence from the human triceps surae. J Appl Physiol 130, 342–354.

Hynngstrom AS, Johnson MD, Miller JF & Heckman CJ (2007). Intrinsic electrical properties of spinal motoneurons vary with joint angle. Nat Neurosci 10, 363–369.

Kuo JJ, Lee RH, Johnson MD, Heckman HM & Heckman CJ (2003). Active Dendritic Integration of Inhibitory Synaptic Inputs in Vivo. J Neurophysiol 90, 3617–3624.

Trajano GS, Seitz LB, Nosaka K & Blazeovich AJ (2014). Can passive stretch inhibit motoneuron facilitation in the human plantar flexors? J Appl Physiol 117, 1486–1492.

Trajano GS, Taylor JL, Orssatto LBR, McNulty CR & Blazeovich AJ (2020). Passive muscle stretching reduces estimates of persistent inward current strength in soleus motor units. J Exp Biol 223, 1–20.

Vandenberk M & Kalmar J (2014). An evaluation of paired motor unit estimates of persistent inward current in human motoneurons. J Neurophysiol 111, 1877-1884.

Referee #2:

This is an original research as well as methodological study involving complicated experimental process and large amount of experimental work. The study focuses on comparison of two methods for estimation of PICs in humans. One method is the well-established paired motor unit (MU) technique and the other the ongoing tendon vibration overlaid by short bursts of neuromuscular electrical stimulation (VibStim). The MU method estimates PICs by calculating the delta frequency (ΔF) of MU firing, while the VibStim method estimates PICs by examining the plantar flexor torque. PICs induced by these two methods

are estimated under two different conditions, the reciprocal inhibition (RI) and whole-body relaxation (WBR), both assumed to reduce PICs during motoneuron firing in humans.

The experimental results showed that ΔF was significantly decreased by both RI and WBR whereas the torques were reduced by WBR but unchanged in RI. Therefore, the paired motor unit (MU) technique can be used potentially to estimate PICs by non-pharmacological neuromodulatory interventions such as WBR. However, it remains unclear if VibStim can be used as a proxy for PIC estimation.

This study could be helpful for us to understand the mechanisms underlying PIC depression in human motoneurons. Especially, non-pharmacological interventions (electrical stimulation or relaxation) might be used to attenuate unwanted PIC-induced muscle contractions triggered by motoneuron hyperexcitability.

Response

We would like to thank the reviewer for taking the time to consider our manuscript.

Major concerns:

1. The major concern on this study is the method of whole-body relaxation (WBR) that the authors used as one intervention to reduce monoamine release of the experimental subjects. Although both arousal state (stress, in particular) and the level of voluntary activity influence the noradrenergic and serotonergic systems, the basis for using WBR as a tool to reduce monoamine release and thus PIC strength remains unclear. In sleep state the activity of neurons in both the pontine and medullary groups are shown to be generally unresponsive to a variety of physiological challenges or stressors in animal models. In WBR state, however, the human subjects are maintained in wakening state and are allowed to participate in the designed experiments. Reduction of their monoaminergic system release is unmeasurable during WBR. Therefore the reliability of the experimental results might be uncertain. Although the authors have briefly discussed WBR in Discussion, the physiological basis for WBR is still not clear. It would be good if the authors could clarify this major issue and give

a stronger, more detailed physiological background to support their usage of WBR in human study.

2. Following the above discussion, the experimental data showed that all VibStim variables remained unaltered in RI and that in VibStim, the T_{vib} and T_{sus} torque were reduced by WBR but other torque and EMG variables remained unchanged. The inconsistent effect in VibStim suggested that WBR might not be used as a proxy for PIC estimation.

Response

We agree with the reviewer that direct measurements of monoaminergic activity are not possible in human studies. Thus, we have made this clearer in our Discussion, explaining to the reader that although we have indirect evidence from previous studies and indirect empirical evidence in this study to support a possible reduction of monoaminergic drive during WBR, this remains speculative. A stronger, more detailed physiological background was now added to the discussion (Pages 46-47, Lines 17-8). Thank you for making this suggestion which, in our view, improved our manuscript.

A spectrum of arousal states exists when awake, with sleepiness and inattention at one end, and stress-induced hypervigilance and panic at the other (Aston-Jones & Bloom, 1981; Ross & Bockstaele, 2021). Activity in locus coeruleus (LC) noradrenergic neurons correlates with arousal state and stress, as directly shown in monkeys (e.g., Rajkowski et al., 1997) and indirectly in humans using magnetic resonance imaging (e.g., Sturm et al., 1999; Naegeli et al., 2018). Modulation of diffusely projecting LC neuronal activity evokes diverse noradrenergic responses, including changes in pupil diameter (Joshi et al., 2016) and heart rate (Gurtu et al., 1984; Ter Horst et al., 1991). Importantly, the dendrites and soma of spinal motoneurons also receive LC input (Proudfit & Clark, 1991). In the present study, the parallel decreases in self-rated stress and heart rate, lower ΔF values, and reduction in two VibStim variables suggest that WBR moderately shifted arousal state, likely decreasing LC activity with a consequent decrease in noradrenergic release onto the motoneurons. Previously, lower peripheral levels of noradrenaline

have been shown following passive, seated relaxation (Teixeira et al., 2005), although future studies are needed to better understand other correlates of LC activity during relaxation such as pupil diameter (Joshi et al., 2016) and brain activity using functional magnetic resonance imaging (Turker et al., 2021). Given that spinal motoneurons also receive diffuse serotonergic innervation from the raphe nuclei (Skagerberg & Björklund, 1985), with serotonin release being triggered by motor activity (Veasey et al., 1995; Wei et al., 2014; Noga et al., 2017), it is also possible that a decrease in activity of other, non-tested muscles during WBR led to an additional decrease in serotonin release. Nonetheless, a reduction in monoaminergic release during WBR remains speculative. ~~Regarding WBR, the reduction in stress scores and heart rate suggests that relaxation was successfully induced.~~

Minor concerns

1. Kernel density estimation (density curves) of the data is represented in Figure 2&4. This process of estimating the probability density function should be briefly described in Methods and the significance of the data should be explained in either text or legends.

Response

We would like to thank the reviewer for these suggestions, and we have added additional descriptions in the methods (statistical analysis) and in the legends of the figures.

Methods (Page 19, Lines 10-13):

For visualisation purposes, kernel density estimations (density curves) of the MU variables were plotted (ggghalves package; Tiedemann, 2020) to depict data distribution. These density curves are a smooth empirical probability density function, and each data point has an equivalent influence on the final distribution.

Figure 2 (Page 26, Lines 4-6):

Kernel density estimation (density curves) of the data is represented on the right by half-violin plots (blue for control and orange for whole-body relaxation). The peak, valleys, and tails of the density curves can be visually compared to see where control and reciprocal inhibition trials were similar or different.

Figure 4 (Page 31, Lines 4-6):

Kernel density estimation (density curves) of the data is represented on the right by half-violin plots (blue for control and orange for whole-body relaxation). The peak, valleys, and tails of the density curves can be visually compared to see where control and whole-body relaxation trials were similar or different.

2. There is no definition of stress score in the Methods, and this should be described. Furthermore, the rationale of the definition and calculation of the scores for this study should be explained.

Response

We think that the expression “stress score” was misleading. When we mentioned “stress score”, we were actually referring to self-rated stress measured with Likert scales. The expression “stress scores” has been replaced by “self-rated stress” throughout the manuscript, including in the methods (Page 18, Lines 2-6), alongside some re-wording:

Self-rated stress during the trials was measured immediately after each trial. Participants answered the question “Please indicate how relaxed or stressed you felt during the trial”. Participants indicated their answer on a seven-point Likert scale, where 1 = Very relaxed, 2 = Relaxed, 3 = Somewhat relaxed, 4 = Neither relaxed or stressed, 5 = Somewhat stressed, 6 = Stressed and 7 = Very stressed. Researchers were blinded to the participants’ answers.

3. PICs are mediated by dihydropyridine (DHP) sensitive L-type calcium currents including HVA and LVA or riluzole (or TTX) sensitive slow-inactivated sodium currents. It would be good if the authors could have some discussion about the component of PICs depressed by RI and WBR in this study.

Response

This is an interesting point. However, we cannot differentiate the relative contributions of the activity from different type of PIC channels during the voluntary, ramp contractions in this study. Thus, we prefer not to add further speculation in this already long paper. As expressed by Binder et al. (2020), the relative contribution of persistent sodium currents vs. persistent calcium currents to the total PIC expressed in motoneurons is likely to vary between behavioural states.

4. Page 38, line 19-20: "Estimates of PICs (ΔF) are reduced by reciprocal inhibition and whole-body" might be stated as "Estimated PICs (ΔF) are reduced by reciprocal inhibition and whole-body relaxation"

Response

We agree with this suggestion, and we have changed the title of the section accordingly.

This study could be helpful for us to understand the mechanisms underlying PIC depression in human motoneurons. Especially, non-pharmacological interventions (electrical stimulation or relaxation) might be used to attenuate unwanted PIC-induced muscle contractions triggered by motoneuron hyperexcitability.

Response

Thank you for this final, positive feedback. We agree that the findings of this study have important translational implications. We are confident it will motivate future

examinations of the effectiveness of certain therapeutical strategies to attenuate motoneuron hyperexcitability.

References

Aston-Jones G, Chen S, Zhu Y & Oshinsky ML (2001). A neural circuit for circadian regulation of arousal. *Nat Neurosci* 4, 732–738.

Binder MD, Powers RK & Heckman CJ (2020). Nonlinear Input-Output Functions of Motoneurons. *Physiology (Bethesda)* 35, 31–39.

Gurtu S, Kamlesh Kumar Pant, Jagdish Narain Sinha & Krishna Prasad Bhargava (1984). An investigation into the mechanism of cardiovascular responses elicited by electrical stimulation of locus coeruleus and subcoeruleus in the cat. *Brain Res* 301, 59–64.

Joshi S, Li Y, Kalwani RM & Gold JI (2016). Relationships between Pupil Diameter and Neuronal Activity in the Locus Coeruleus, Colliculi, and Cingulate Cortex. *Neuron* 89, 221–234.

Proudfit HK & Clark FM (1991). The projections of locus coeruleus neurons to the spinal cord. *Prog Brain Res* 88, 123–141.

Naegeli C, Zeffiro T, Piccirelli M, Jaillard A, Weilenmann A, Hassanpour K, Schick M, Rufer M, Orr SP & Mueller-Pfeiffer C (2018). Locus Coeruleus Activity Mediates Hyperresponsiveness in Posttraumatic Stress Disorder. *Biol Psychiatry* 83, 254–262.

Noga BR, Turkson RP, Xie S, Taberner A, Pinzon A & Hentall ID (2017). Monoamine release in the cat lumbar spinal cord during fictive locomotion evoked by the mesencephalic locomotor region. *Front Neural Circuits* 11, 1–24.

Rajkowski J, Kubiak P, Ivanova S & Aston-Jones G (1997). State-Related Activity, Reactivity of Locus Ceruleus Neurons in Behaving Monkeys. *Adv Pharmacol* 42, 740–744.

Ross JA & Van Bockstaele EJ (2021). The Locus Coeruleus- Norepinephrine System in Stress and Arousal: Unraveling Historical, Current, and Future Perspectives. Front Psychiatry 11, 1–23.

Skagerberg G & Björklund A (1985). Topographic principles in the spinal projections of serotonergic and non-serotonergic brainstem neurons in the rat. Neuroscience 15, 445–480.

Sturm W, De Simone A, Krause BJ, Specht K, Hesselmann V, Radermacher I, Herzog H, Tellmann L, Müller-Gärtner HW & Willmes K (1999). Functional anatomy of intrinsic alertness: Evidence for a fronto-parietal-thalamic-brainstem network in the right hemisphere. Neuropsychologia 37, 797–805.

Teixeira J, Martin D, Prendiville O & Glover V (2005). The effects of acute relaxation on indices of anxiety during pregnancy. J Psychosom Obstet Gynecol 26, 271–276.

Ter Horst GJ, Toes GJ & van Willigen JD (1991). Locus coeruleus projections to the dorsal motor vagus nucleus in the rat. Neuroscience 45, 153–160.

Turker HB, Riley E, Luh WM, Colcombe SJ & Swallow KM (2021). Estimates of locus coeruleus function with functional magnetic resonance imaging are influenced by localization approaches and the use of multi-echo data. Neuroimage 236, 118047.

Veasey SC, Fornal CA, Metzler CW & Jacobs BL (1995). Response of serotonergic caudal raphe neurons in relation to specific motor activities in freely moving cats. J Neurosci 15, 5346–5359.

Wei K, Glaser JI, Deng L, Thompson CK, Stevenson IH, Wang Q, Hornby TG, Heckman CJ & Kording KP (2014). Serotonin Affects Movement Gain Control in the Spinal Cord. J Neurosci 34, 12690–12700.

Dear Dr Mesquita,

Re: JP-RP-2022-282765R1 "Effects of reciprocal inhibition and whole-body relaxation on persistent inward currents estimated by two different methods" by Ricardo N. O. Mesquita, Janet L Taylor, Gabriel S. Trajano, Jakob Škarabot, Ales Holobar, Basílio A. M. Gonçalves, and Anthony Blazevich

I am pleased to tell you that your paper has been accepted for publication in The Journal of Physiology.

NEW POLICY: In order to improve the transparency of its peer review process The Journal of Physiology publishes online as supporting information the peer review history of all articles accepted for publication. Readers will have access to decision letters, including all Editors' comments and referee reports, for each version of the manuscript and any author responses to peer review comments. Referees can decide whether or not they wish to be named on the peer review history document.

The last Word version of the paper submitted will be used by the Production Editors to prepare your proof. When this is ready you will receive an email containing a link to Wiley's Online Proofing System. The proof should be checked and corrected as quickly as possible.

Authors should note that it is too late at this point to offer corrections prior to proofing. The accepted version will be published online, ahead of the copy edited and typeset version being made available. Major corrections at proof stage, such as changes to figures, will be referred to the Reviewing Editor for approval before they can be incorporated. Only minor changes, such as to style and consistency, should be made a proof stage. Changes that need to be made after proof stage will usually require a formal correction notice.

All queries at proof stage should be sent to TJP@wiley.com

Are you on Twitter? Once your paper is online, why not share your achievement with your followers. Please tag The Journal (@jphysiol) in any tweets and we will share your accepted paper with our 23,000+ followers!

Yours sincerely,

Richard Carson
Senior Editor
The Journal of Physiology

P.S. - You can help your research get the attention it deserves! Check out Wiley's free Promotion Guide for best-practice recommendations for promoting your work at www.wileyauthors.com/eoo/guide. And learn more about Wiley Editing Services which offers professional video, design, and writing services to create shareable video abstracts, infographics, conference posters, lay summaries, and research news stories for your research at www.wileyauthors.com/eoo/promotion.

*** IMPORTANT NOTICE ABOUT OPEN ACCESS ***

Information about Open Access policies can be found here <https://physoc.onlinelibrary.wiley.com/hub/access-policies>

To assist authors whose funding agencies mandate public access to published research findings sooner than 12 months after publication The Journal of Physiology allows authors to pay an open access (OA) fee to have their papers made freely available immediately on publication.

You will receive an email from Wiley with details on how to register or log-in to Wiley Authors Services where you will be able to place an OnlineOpen order.

You can check if your funder or institution has a Wiley Open Access Account here <https://authorservices.wiley.com/author-resources/Journal-Authors/licensing-and-open-access/open-access/author-compliance-tool.html>

Your article will be made Open Access upon publication, or as soon as payment is received.

If you wish to put your paper on an OA website such as PMC or UKPMC or your institutional repository within 12 months of publication you must pay the open access fee, which covers the cost of publication.

OnlineOpen articles are deposited in PubMed Central (PMC) and PMC mirror sites. Authors of OnlineOpen articles are permitted to post the final, published PDF of their article on a website, institutional repository, or other free public server, immediately on publication.

Note to NIH-funded authors: The Journal of Physiology is published on PMC 12 months after publication, NIH-funded

authors DO NOT NEED to pay to publish and DO NOT NEED to post their accepted papers on PMC.

REQUIRED ITEM

The reference number for ethics approval should be provided.

EDITOR COMMENTS

The revised manuscript has well addressed the concerns raised by the reviewers, and demonstrated the effects of reciprocal inhibition and whole-body relaxation on PICs, suggesting the potential of clinical interventions using neuromodulation or relaxation. Please provide the reference number for ethics approval in the revised manuscript.

REFEREE COMMENTS

Referee #1:

The authors have provided a thoughtful and effective revision. I have no further concerns. The results are highly interesting and the experiments rigorously performed and analyzed.

Referee #2:

As to my major concerns on WBR and the inconsistent effect in VibStim, the authors have added a detailed physiological background to the discussion, which clarifies the confusion points on the reliability of their experimental results. Also, the authors have well addressed all the issues about my minor concerns. I have no more concerns on this revised manuscript.

1st Confidential Review

22-Mar-2022

REQUIRED ITEM

The reference number for ethics approval should be provided.

EDITOR COMMENTS

The revised manuscript has well addressed the concerns raised by the reviewers, and demonstrated the effects of reciprocal inhibition and whole-body relaxation on PICs, suggesting the potential of clinical interventions using neuromodulation or relaxation. Please provide the reference number for ethics approval in the revised manuscript.

REFEREE COMMENTS

Referee #1:

The authors have provided a thoughtful and effective revision. I have no further concerns. The results are highly interesting and the experiments rigorously performed and analyzed.

Referee #2:

As to my major concerns on WBR and the inconsistent effect in VibStim, the authors have added a detailed physiological background to the discussion, which clarifies the confusion points on the reliability of their experimental results. Also, the authors have well addressed all the issues about my minor concerns. I have no more concerns on this revised manuscript.

Response: We would like to thank the editors and the reviewers for their comments. We look forward to seeing our article published in the Journal of Physiology and disseminate our research findings with the scientific community. We have added the reference number of the ethics approval, as requested.

Dear Mr Mesquita,

Re: JP-RP-2022-282765R2 "Effects of reciprocal inhibition and whole-body relaxation on persistent inward currents estimated by two different methods" by Ricardo N. O. Mesquita, Janet L Taylor, Gabriel S. Trajano, Jakob Škarabot, Ales Holobar, Basílio A. M. Gonçalves, and Anthony Blazevich

Thank you for submitting your manuscript to The Journal of Physiology. It has been assessed by a Reviewing Editor and I am pleased to tell you that it is considered to be acceptable for publication following satisfactory revision.

The reports are copied at the end of this email. Please address all of the points and incorporate all requested revisions, or explain in your Response to Referees why a change has not been made.

NEW POLICY: In order to improve the transparency of its peer review process The Journal of Physiology publishes online as supporting information the peer review history of all articles accepted for publication. Readers will have access to decision letters, including all Editors' comments and referee reports, for each version of the manuscript and any author responses to peer review comments. Referees can decide whether or not they wish to be named on the peer review history document.

Authors are asked to use The Journal's premium BioRender (<https://biorender.com/>) account to create/redraw their Abstract Figures. Information on how to access The Journal's premium BioRender account is here:

<https://physoc.onlinelibrary.wiley.com/journal/14697793/biorender-access> and authors are expected to use this service. This will enable Authors to download high-resolution versions of their figures. The link provided should only be used for the purposes of this submission. Authors will be charged for figures created on this premium BioRender account if they are not related to this manuscript submission.

I hope you will find the comments helpful and have no difficulty returning your revisions within 4 weeks.

Your revised manuscript should be submitted online using the links in Author Tasks: Link Not Available.

Any image files uploaded with the previous version are retained on the system. Please ensure you replace or remove all files that have been revised.

REVISION CHECKLIST:

- Article file, including any tables and figure legends, must be in an editable format (eg Word)
- Abstract figure file (see above)
- Statistical Summary Document
- Upload each figure as a separate high quality file
- Upload a full Response to Referees, including a response to any Senior and Reviewing Editor Comments;
- Upload a copy of the manuscript with the changes highlighted.

- A potential 'Cover Art' file for consideration as the Issue's cover image;
- Appropriate Supporting Information (Video, audio or data set https://jp.msubmit.net/cgi-bin/main.plex?form_type=display_requirements#supp).

To create your 'Response to Referees' copy all the reports, including any comments from the Senior and Reviewing Editors, into a Word, or similar, file and respond to each point in colour or CAPITALS and upload this when you submit your revision.

I look forward to receiving your revised submission.

If you have any queries please reply to this email and staff will be happy to assist.

Yours sincerely,

Richard Carson
Senior Editor
The Journal of Physiology

REQUIRED ITEMS:

- Your manuscript must include a complete Additional Information section

- A Data Availability Statement is required for all papers reporting original data. This must be in the Additional Information section of the manuscript itself. It must have the paragraph heading "Data Availability Statement". All data supporting the results in the paper must be either: in the paper itself; uploaded as Supporting Information for Online Publication; or archived in an appropriate public repository. The statement needs to describe the availability or the absence of shared data. Authors must include in their Statement: a link to the repository they have used, or a statement that it is available as Supporting Information; reference the data in the appropriate sections(s) of their manuscript; and cite the data they have shared in the References section. Whenever possible the scripts and other artefacts used to generate the analyses presented in the paper should also be publicly archived. If sharing data compromises ethical standards or legal requirements then authors are not expected to share it, but must note this in their Statement. For more information, see our Statistics Policy.

END OF COMMENTS

2nd Confidential Review

04-Apr-2022

REQUIRED ITEMS:

- Your manuscript must include a complete Additional Information section

- A Data Availability Statement is required for all papers reporting original data. This must be in the Additional Information section of the manuscript itself. It must have the paragraph heading "Data Availability Statement". All data supporting the results in the paper must be either: in the paper itself; uploaded as Supporting Information for Online Publication; or archived in an appropriate public repository. The statement needs to describe the availability or the absence of shared data. Authors must include in their Statement: a link to the repository they have used, or a statement that it is available as Supporting Information; reference the data in the appropriate sections(s) of their manuscript; and cite the data they have shared in the References section. Whenever possible the scripts and other artefacts used to generate the analyses presented in the paper should also be publicly archived. If sharing data compromises ethical standards or legal requirements then authors are not expected to share it, but must note this in their Statement. For more information, see our Statistics Policy.

END OF COMMENTS

Response: We have changed our Data Availability Statement and made individual data that support the findings of this study available as a supporting information file.

Dear Dr Mesquita,

Re: JP-RP-2022-282765R3 "Effects of reciprocal inhibition and whole-body relaxation on persistent inward currents estimated by two different methods" by Ricardo N. O. Mesquita, Janet L Taylor, Gabriel S. Trajano, Jakob Škarabot, Ales Holobar, Basílio A. M. Gonçalves, and Anthony Blazevich

I am pleased to tell you that your paper has been accepted for publication in The Journal of Physiology.

NEW POLICY: In order to improve the transparency of its peer review process The Journal of Physiology publishes online as supporting information the peer review history of all articles accepted for publication. Readers will have access to decision letters, including all Editors' comments and referee reports, for each version of the manuscript and any author responses to peer review comments. Referees can decide whether or not they wish to be named on the peer review history document.

The last Word version of the paper submitted will be used by the Production Editors to prepare your proof. When this is ready you will receive an email containing a link to Wiley's Online Proofing System. The proof should be checked and corrected as quickly as possible.

Authors should note that it is too late at this point to offer corrections prior to proofing. The accepted version will be published online, ahead of the copy edited and typeset version being made available. Major corrections at proof stage, such as changes to figures, will be referred to the Reviewing Editor for approval before they can be incorporated. Only minor changes, such as to style and consistency, should be made a proof stage. Changes that need to be made after proof stage will usually require a formal correction notice.

All queries at proof stage should be sent to TJP@wiley.com.

Are you on Twitter? Once your paper is online, why not share your achievement with your followers. Please tag The Journal (@jphysiol) in any tweets and we will share your accepted paper with our 23,000+ followers!

Yours sincerely,

Richard Carson
Senior Editor
The Journal of Physiology

P.S. - You can help your research get the attention it deserves! Check out Wiley's free Promotion Guide for best-practice recommendations for promoting your work at www.wileyauthors.com/eeo/guide. And learn more about Wiley Editing Services which offers professional video, design, and writing services to create shareable video abstracts, infographics, conference posters, lay summaries, and research news stories for your research at www.wileyauthors.com/eeo/promotion.

*** IMPORTANT NOTICE ABOUT OPEN ACCESS ***

Information about Open Access policies can be found here <https://physoc.onlinelibrary.wiley.com/hub/access-policies>

To assist authors whose funding agencies mandate public access to published research findings sooner than 12 months after publication The Journal of Physiology allows authors to pay an open access (OA) fee to have their papers made freely available immediately on publication.

You will receive an email from Wiley with details on how to register or log-in to Wiley Authors Services where you will be able to place an OnlineOpen order.

You can check if your funder or institution has a Wiley Open Access Account here <https://authorservices.wiley.com/author-resources/Journal-Authors/licensing-and-open-access/open-access/author-compliance-tool.html>

Your article will be made Open Access upon publication, or as soon as payment is received.

If you wish to put your paper on an OA website such as PMC or UKPMC or your institutional repository within 12 months of publication you must pay the open access fee, which covers the cost of publication.

OnlineOpen articles are deposited in PubMed Central (PMC) and PMC mirror sites. Authors of OnlineOpen articles are permitted to post the final, published PDF of their article on a website, institutional repository, or other free public server, immediately on publication.

Note to NIH-funded authors: The Journal of Physiology is published on PMC 12 months after publication, NIH-funded

authors DO NOT NEED to pay to publish and DO NOT NEED to post their accepted papers on PMC.

3rd Confidential Review

11-Apr-2022